# HYPERBOLIC FINE-TUNING FOR LARGE LANGUAGE MODELS

## ABSTRACT

Large language models (LLMs) have demonstrated remarkable performance on various tasks. However, it remains an open question whether the default Euclidean space is the most suitable choice for embedding tokens in LLMs. In this study, we first investigate the non-Euclidean characteristics of LLMs. Our findings reveal that token frequency follows a power-law distribution, with high-frequency tokens clustering near the origin and low-frequency tokens positioned farther away. Additionally, token embeddings exhibit a high degree of hyperbolicity, indicating a latent tree-like structure in the embedding space. Building on the observation, we propose to efficiently fine-tune LLMs in hyperbolic space to better exploit the underlying complex structures. However, we found that this fine-tuning in hyperbolic space cannot be achieved with naive application of exponential and logarithmic maps, when the embedding and weight matrices both reside in Euclidean space. To address this technique issue, we introduce a new method called hyperbolic low-rank efficient fine-tuning, HypLoRA, that performs low-rank adaptation directly on the hyperbolic manifold, avoiding the cancellation effect caused by the exponential and logarithmic maps, thus preserving the hyperbolic modeling capabilities. Through extensive experiments, we demonstrate that HypLoRA significantly enhances the performance of LLMs on reasoning tasks, particularly for complex reasoning problems. In particular, HypLoRA improves the performance in the complex AQuA dataset by up to 13.0%, showcasing its effectiveness in handling complex reasoning challenges.

## 1 INTRODUCTION

Large language models (LLMs) such as GPT-4 (Achiam et al., 2023), LLaMA (Touvron et al., 2023), and Gemma (Gemma Team, 2024) have demonstrated remarkable capabilities in understanding and generating human-like text (Qin et al., 2023; Shen et al., 2024). Despite their impressive capabilities, these models often rely on Euclidean geometry for learning text representations, which may not always be the complex, hierarchical nature of real-world data structures (Bronstein et al., 2017; Bachmann et al., 2020). For example, in language, words are often organized into categories that reflect varying levels of abstraction. These relationships naturally form a tree-like structure, where general or abstract concepts, such as "fruits," sit at the top of the hierarchy, while more specific or concrete terms, like "apples" or "bananas," reside at the lower levels. Representing such structures effectively is crucial for understanding the semantics of language in LLMs.

Recent advancements suggest that non-Euclidean geometries, particularly hyperbolic spaces, offer promising alternatives for modeling hierarchical data. Hyperbolic space, distinguished by its negative curvature, is especially well-suited for representing tree-like data due to its exponential volume growth, enabling efficient embeddings of hierarchies (Nickel & Kiela, 2017; 2018; Ganea et al., 2018a; Khrulkov et al., 2020; Cetin et al., 2022). However, a significant research gap remains: existing works have not attempted to study LLM embeddings in the context of non-Euclidean geometry.

**Proposed Analysis Framework** In this work, we first delve deep into how LLMs interact with token embeddings and explore to what extent these embeddings exhibit non-Euclidean characteristics. We approach this from both a global and local perspective. At the global level, we analyze the overall distribution of tokens by frequency, examining how these frequency maps are distributed across the embedding space. At the local level, we measure the hyperbolicity (Borassi et al., 2015; Kennedy

et al., 2013) of the metric space spanned by each input prompt, where the embedding hyperbolicity serves as a proxy to assess the similarity of the underlying embedding structure to a tree structure.

Our analysis in Section 4 reveals several key insights. First, token frequency follows a power-law distribution, as shown in Figure 2 (left). This distribution, where a small set of tokens appears frequently while most are rare, suggests an implicit hierarchy similar to a branching tree (Krioukov et al., 2010). High-frequency tokens (abstract concepts) tend to be located near the origin of the embedding space, while low-frequency tokens (specific terms) are farther away, as depicted in Figure 2 (right) and Table 1. Furthermore, our investigation of hyperbolicity ($\delta$ values) in Table 2 demonstrates that LLM token embeddings exhibit significant tree-like properties.

Based on our findings above, a natural consideration is to develop hyperbolic LLMs that explicitly incorporate hyperbolic inductive bias, as shown in Figure 1. However, training LLMs

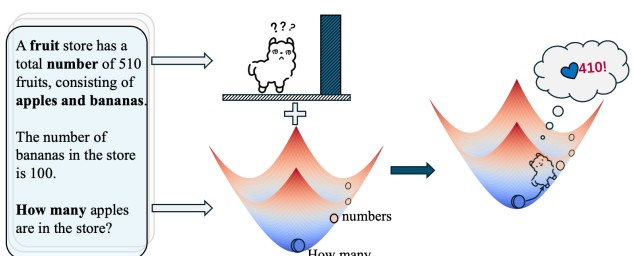

Figure 1: Illustration of a reasoning task with hyperbolic geometry on LLMs. The figure shows that a fruit store has 510 fruits (apples and bananas), and knowing that there are 100 bananas, we determine the number of apples to be 410. In token embedding, frequent and abstract tokens (like, "fruit, how many, and numbers") are represented closer to the origin, while specific and less frequent tokens (like "apples, bananas, numbers, 510, 110") appear further away, creating a tree-like structure. By this structure, hyperbolic space enables LLMs to map hierarchical relationships more efficiently, preserving the inherent structure of tokens, understanding the semantic meaning, and facilitating accurate reasoning.

from scratch can be resource-intensive (Loshchilov & Hutter, 2017; Rajbhandari et al., 2020). As a more resource-efficient alternative, we propose to build the first low-rank adaptation method in hyperbolic space. This approach is particularly advantageous given that existing LLMs are all Euclidean, and not all downstream tasks require hyperbolic geometry in their fine-tuning. Through employing hyperbolic adapters for specific tasks on an Euclidean foundation model, we can leverage the benefits of both geometries while maintaining computational efficiency.

**Challenges** Adapting LLMs in non-Euclidean embedding spaces with classic techniques, *i.e.* applying exponential and logarithmic maps with tangent space (Chami et al., 2019; Ganea et al., 2018b; Yang et al., 2022c) for weight adaptation is problematic in this case. This approach fails to fully capture the hyperbolic geometry, as the exponential and logarithmic maps are mutually inverse and can be canceled with consecutive operations. Consequently, the inherent properties of the hyperbolic space are not effectively preserved, limiting the potential benefits of incorporating non-Euclidean geometries into the adaptation process.

**Hyperbolic Fine-tuning** To address this limitation, we introduce HypLoRA to operate low-rank adaptation directly on the hyperbolic manifold without transformation to the tangent space, thus preserving hyperbolic modeling capabilities and counteracting the reduction[1]. HypLoRA integrates hyperbolic geometry into existing LLMs, introducing implicitly high-order interaction and considering the token hierarchies, enabling them to benefit from hyperbolic characteristics while minimizing additional computational costs.

To summarize, our main contributions are threefold: (1) We conduct a comprehensive investigation into the hyperbolic characteristics of token embeddings in LLMs, revealing their inherent tree-like structure and strong hyperbolic properties. (2) We propose HypLoRA, a parameter-efficient fine-tuning method that integrates hyperbolic geometry into LLMs while preserving hyperbolic modeling capabilities. We show that HypLoRA better understands complex reasoning tasks by implicitly incorporating high-order interactions and token norms, achieving improvements of up to 13% on the challenging AQuA dataset. Our work opens new avenues for exploring the role of geometry in LLMs and provides insights for developing geometrically informed models for reasoning tasks.

---

[1]Code is available at https://anonymous.4open.science/r/HypLLMs-DD7A

## 2 RELATED WORK

**Hyperbolic Representation Learning and Deep Learning** Hyperbolic geometry has been successfully applied to various neural network architectures and models (Yang et al., 2022b; Mettes et al., 2023; Peng et al., 2021), including shallow hyperbolic neural networks (Ganea et al., 2018a;b; Chen et al., 2021; Shimizu et al., 2020), hyperbolic CNNs (Bdeir et al., 2023; van Spengler et al., 2023), and hyperbolic attention networks or Transformers (Gulcehre et al., 2018; Chen et al., 2021; Shimizu et al., 2020; Yang et al., 2024). These models leverage the inductive biases of hyperbolic geometry to achieve remarkable performance on various tasks and applications (Chami et al., 2019; Yang et al., 2022a; Sun et al., 2021; Khrulkov et al., 2020; Cetin et al., 2022; Weng et al., 2021; Xiong et al., 2022; Yang et al., 2021). However, training LLMs from scratch remains computationally expensive (Kochurov et al., 2020; Smith, 2014). The computational complexity increases further when considering Riemannian optimization (Kochurov et al., 2020; Smith, 2014; Bécigneul & Ganea, 2018) and additional hyperbolic operations, like Möbius addition.

**Parameter Efficient Fine Tuning (PEFT) and LoRAs** Fine-tuning LLMs (Foundation, 2022; 2023; Touvron et al., 2023) for downstream tasks poses significant challenges due to their massive number of parameters. To address this issue, PEFT methods have been proposed, which aim to train a small subset of parameters while achieving better performance compared to full fine-tuning. PEFT methods can be broadly categorized into prompt-based methods (Lester et al., 2021; Li & Liang, 2021; Qin et al., 2021), adapter-based methods (Houlsby et al., 2019; Zhu et al., 2021),and reparameterization-based methods (Hu et al., 2021; Aghajanyan et al., 2020; Edalati et al., 2022). Among these, LoRA (Hu et al., 2021) as the reparameterization-based method, has gained significant attention due to its simplicity, effectiveness, and compatibility with existing model architectures. Variants of LoRA, such as LoRA+(Hayou et al., 2024), DoRA (Liu et al., 2024), AdaLoRA (Zhang et al., 2023), have been proposed to improve its performance and efficiency. Recent research has also investigated ensembles of multiple LoRAs (Wang et al., 2023; Ren et al., 2024), and quantization techniques (Dettmers et al., 2024; Xu et al., 2023; Li et al., 2023). Despite these advances, existing methods operate within Euclidean space, ignoring the underlying structure represented by LLMs. The proposed method is as a foundational algorithm, potentially combined with various LoRA variants, to exploit their complementary strengths and achieve superior performance.

## 3 PRELIMINARY

This section introduces the concepts utilized in our study, including the LoRA adapter, the Lorentz model of hyperbolic geometry, hyperbolic linear transformations, and the concept of hyperbolicity.

**LoRA Adapter** The LoRA adapter offers an efficient approach for modifying large LLMs with minimal computational overhead. Instead of retraining the entire model, LoRA focuses on adjusting specific components within the model's architecture to transform an input $\mathbf{x}$ into an output $\mathbf{z}$. In practice, LoRA targets the weight matrices found in each Transformer layer of an LLM. Typically, the weight $W$ of the Transformer, which resides in the dimensions $\mathbb{R}^{d \times k}$, is adapted through a low-rank approximation. This is achieved by introducing an additional term, $\Delta W$, to the original weight matrix:

$$\mathbf{z} = W_{\text{LoRA}}(\mathbf{x}) = W\mathbf{x} + \Delta W\mathbf{x} = W\mathbf{x} + BA\mathbf{x}. \tag{1}$$

Here, $B \in \mathbb{R}^{d \times r}$ and $A \in \mathbb{R}^{r \times k}$ represent two smaller, learnable matrices where $r$—the rank of these matrices—is significantly less than either $d$ or $k$. This design choice ensures that $r \ll \min(d, k)$, thereby reducing the complexity of the model adaptation. During the fine-tuning process, only the matrices $A$ and $B$ are adjusted, while the pre-existing weights $W$ are kept frozen. This method significantly decreases the number of parameters that need to be trained, from $dk$ to $(d + k)r$, enhancing the efficiency of the fine-tuning process. As a result, LoRA enables the targeted adaptation of LLMs, allowing them to transform an input $\mathbf{x}$ into an output $\mathbf{z}$ while maintaining high performance and adapting to new tasks or datasets with a fraction of the computational resources typically required.

**Hyperbolic Geometry** Unlike the flat Euclidean geometry, hyperbolic geometry is characterized by a constant negative curvature. We utilize the Lorentz model, also known as the hyperboloid model, for our study due to its ability to effectively capture hierarchical structures and maintain numerical stability (Nickel & Kiela, 2018; Chen et al., 2021). The Lorentz model in $n$ dimensions with curvature

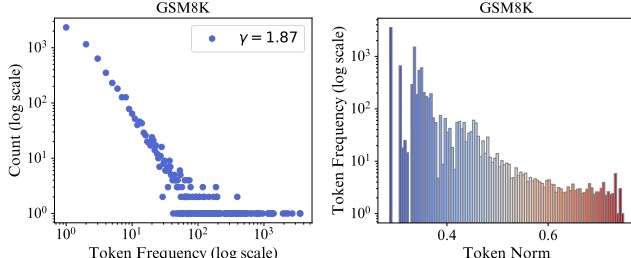

Figure 2: Token frequency distribution (left) and token frequency vs. norm (bottom row) of GSM8K datast in LLaMA3. The top row shows the power-law distribution of token frequencies with the decay rate ($\gamma$) annotated for each dataset. The bottom row illustrates the relationship between token frequency and token norm, binned and colored by frequency, where higher token norms correspond to lower frequencies. For more data illustration, please refer to Appendix A.

$-1/K(K > 0)$ is defined as:

$$\mathcal{L}_K^n = \{\mathbf{x} \in \mathbb{R}^{n+1} : \langle \mathbf{x}, \mathbf{x} \rangle_{\mathcal{L}} = -K, x_0 > 0\}, \tag{2}$$

where $\langle \cdot, \cdot \rangle_{\mathcal{L}}$ is the Lorentzian inner product, given by: $\langle \mathbf{x}, \mathbf{y} \rangle_{\mathcal{L}} = -x_0 y_0 + \sum_{i=1}^n \mathbf{x}_i \mathbf{y}_i$.

**Tangent Space** In the Lorentz model $\mathcal{L}_K^n$, the tangent space at a point $\mathbf{x}$ is denoted as $\mathcal{T}_{\mathbf{x}} \mathcal{L}_K^n$. It is defined as the set of all vectors $\mathbf{u}$ that are orthogonal to $\mathbf{x}$ under the Lorentzian inner product:

$$\mathcal{T}_{\mathbf{x}} \mathcal{L}_K^n := \{\mathbf{u} \in \mathbb{R}^{n+1} : \langle \mathbf{u}, \mathbf{x} \rangle_{\mathcal{L}} = 0\}. \tag{3}$$

To facilitate projection between the hyperboloid and its tangent spaces, we utilize two critical mappings: the exponential and logarithmic maps. The *exponential map* at $\mathbf{x}$, denoted $\exp_{\mathbf{x}}^K$, projects a vector from the tangent space $\mathcal{T}_{\mathbf{x}} \mathcal{L}_K^n$ back onto the hyperboloid. Conversely, the *logarithmic map*, denoted $\log_{\mathbf{x}}^K$, maps a point on the hyperboloid to the tangent space at $\mathbf{x}$. The relevant formulas are given in Appendix C.

## 4 INVESTIGATION

In this section, we present an in-depth investigation of token embeddings in LLMs from both global and local perspectives. Our goal is to uncover the geometric structures underlying pretrained token representations, specifically examining the global distribution of token frequencies and their spatial arrangement, as well as the local hyperbolicity of token embeddings across various datasets.

### 4.1 GLOBAL TOKEN STATISTICS

We begin by investigating the global distribution of token frequencies in the context of arithmetic reasoning datasets, focusing on datasets such as GSM8K (Cobbe et al., 2021), AQuA (Ling et al., 2017), MAWPS (Koncel-Kedziorski et al., 2016), and SVAMP (Patel et al., 2021). We also provide a broader analysis across different types of datasets and LLMs in Appendix A.

Figure 2 (left) presents the distribution of token frequencies, with a power-law exponent of approximately $\gamma \approx 1.9$, as estimated by the Powerlaw Package (Alstott et al., 2014). In such distributions, the exponent $\gamma$ controls how quickly token frequencies decline: smaller values of $\gamma$ (closer to 1) indicate a more gradual decay where frequent tokens dominate, while larger values signify a sharper decline, with most tokens being rare.

This power-law behavior aligns with the hierarchical nature of language. High-frequency tokens often correspond to more abstract or general concepts, while low-frequency tokens represent specific or rare terms. This distribution naturally suggests a hierarchical organization of the token space, where general concepts serve as the "roots" and specific terms "branch out" as we move through the hierarchy.

To better understand the relationship between token frequency and their spatial arrangement within the embedding space, we calculate the average token frequency as a function of their distance from the origin. The results are shown in Figure 2 (right), indicating that more frequent tokens tend to have smaller norms and vice versa. Table 1 provides representative with different norm ranges within the embedding space. Tokens with smaller norms (ranging from 0.3 to 0.4) include common function words like "to,"

Table 1: Tokens Across Norm Ranges

| Ranges | Representative Tokens |
|---|---|
| $0.3 \sim 0.4$ | to, in, have, that, and, is, for |
| $0.4 \sim 0.5$ | how, much, many, time, cost |
| $0.5 \sim 0.6$ | animals, fruit, numbers, items, colors |
| $0.6 \sim 0.7$ | dog, cow, puppies, apple, bananas, hours, second, minute, dollars, shoes, purple |

"in," and "that," which tend to occur frequently in language. As the norm increases, the tokens become more specific, with ranges from 0.4 to 0.5 including terms like "how," "much," and "time," and further up, with norms between 0.5 and 0.6, featuring more concrete nouns like "animals," "fruit," and "numbers." Tokens with the highest norms, between 0.6 and 0.7, are even more detailed, referring to items such as "dog," "apple," and "dollars."

These findings suggest that the spatial organization of token embeddings reflects the inherent hierarchical relationships in language, supporting the hypothesis that LLMs exhibit a tree-like structure in their token embeddings, with spatial positioning aligned with token frequency and specificity.

### 4.2 $\delta$-HYPERBOLICITY OF TOKEN EMBEDDINGS

To rigorously quantify the hierarchical nature of token embeddings, we examine the $\delta$-hyperbolicity of space spanned by the token embedding. $\delta$-Hyperbolicity, introduced by Gromov (Gromov, 1987), is a measure that captures the degree to which a metric space deviates from an exact tree structure. Lower values of $\delta$ imply a space more similar to a perfect tree, while higher values indicate deviation from a tree-like structure. A brief explanation of $\delta$-hyperbolicity can be found on Wikipedia[2].

We compute $\delta$-hyperbolicity using the four-point condition, which compares the Gromov products between any four points $a$, $b$, $c$, and $w$ in the metric space. Specifically, the hyperbolicity is defined as:

$$[a, c]_w \geq \min([a, b]_w, [b, c]_w) - \delta, \tag{4}$$

where the Gromov product $[a, b]_w$ is:

$$[a, b]_w = \frac{1}{2}(d(a, w) + d(b, w) - d(a, b)). \tag{5}$$

To measure the hyperbolicity of token embeddings, we apply this algorithm to various open-source LLMs. Following the methodologies proposed by Khrulkov et al. (Khrulkov et al., 2020) and Cetin et al. (Cetin et al., 2022), we estimate $\delta$-hyperbolicity using the efficient algorithm introduced by Fournier et al. (Fournier et al., 2015). To ensure scale invariance, we normalize $\delta$ by the diameter of the embedding space, $\mathrm{diam}(X)$, yielding a relative measure: $\delta_{rel} = \frac{2\delta}{\mathrm{diam}(X)}$. This relative measure ranges from 0 to 1, with values closer to 0 indicating a highly hyperbolic (tree-like) structure, and values near 1 indicating a non-hyperbolic, flat structure. Following previous works (Khrulkov et al., 2020), we employ Euclidean distance as a measure of the shortest distance. To further validate the correctness of this approach, we generate a series of random graphs with predefined hyperbolicity, embed them using a graph neural network (GNN), and then compute the hyperbolicity in Euclidean space. Details of this process are provided in Appendix B. Our experiments reveal a positive correlation between the hyperbolicity of the embeddings and the original graphs. Consequently, we utilize this method as a proxy for estimating the hyperbolicity of token embeddings.

In our analysis, we calculate hyperbolicity at the prompt level, treating each token within a prompt as a point in the metric space spanned by the embeddings. By averaging the hyperbolicity across all prompts, we assess the overall hyperbolic structure of token embeddings in each dataset. Our results, as shown in Table 2, reveal that token embeddings exhibit significant hyperbolicity, suggesting that the embedding space has a strong tree-like structure. This observation further corroborates our findings from the global token statistics, where the arrangement of tokens in the embedding space mirrors hierarchical relationships seen in language data.

---

[2]https://en.wikipedia.org/wiki/Hyperbolic_metric_space

Table 2: Comparison of $\delta$-Hyperbolicity across various metric spaces and datasets. The left table provides reference values for baseline metric spaces, allowing for a clearer interpretation of hyperbolicity in the analyzed datasets in the right table.

| Metric Space | Hyperbolicity($\delta$) |
|---|---|
| Sphere Space | $0.99 \pm 0.01$ |
| Random Graph | $0.62 \pm 0.34$ |
| PubMed Graph | $0.40 \pm 0.45$ |
| Scale-free Graph | $0.00$ |
| Tree Graph | $0.00$ |

| Hyperbolicity($\delta$) | MAWPS | SVAMP | GSM8K | AQuA |
|---|---|---|---|---|
| LLaMA-7B | $0.08 \pm 0.02$ | $0.09 \pm 0.01$ | $0.10 \pm 0.01$ | $0.10 \pm 0.01$ |
| LLaMA-13B | $0.08 \pm 0.01$ | $0.09 \pm 0.01$ | $0.09 \pm 0.01$ | $0.10 \pm 0.01$ |
| Gemma-7B | $0.11 \pm 0.01$ | $0.11 \pm 0.01$ | $0.11 \pm 0.01$ | $0.12 \pm 0.01$ |
| LLaMA3-8B | $0.06 \pm 0.01$ | $0.07 \pm 0.01$ | $0.07 \pm 0.01$ | $0.08 \pm 0.01$ |
| Average | $0.08 \pm 0.01$ | $0.09 \pm 0.01$ | $0.09 \pm 0.01$ | $0.10 \pm 0.01$ |

**Conclusion of investigation** Through these analyses, we demonstrate that token embeddings in LLMs exhibit a power-law frequency distribution and significant hyperbolicity, both of which reflect a tree-like hierarchical structure. This understanding not only sheds light on the geometric nature of token embeddings but also motivates the development of methods that better capture and preserve these underlying geometric properties.

## 5 HYPERBOLIC FINE-TUNING FOR LLMS

The core technique in the LoRA adapter involves linear transformations. One of the primary methods for implementing linear transformations on the Lorentz model of hyperbolic geometry (Ganea et al., 2018b; Chami et al., 2019) is based on the tangent space when considering the learnable weights are in Euclidean. Given a hyperbolic vector $\mathbf{x}^H$ and a transformation matrix $W$, this method first maps $\mathbf{x}^H$ to the tangent space at a local reference point, typically the origin, using the logarithmic map. The matrix $W$ is then applied within this tangent space, resulting in:

$$W \otimes \mathbf{x}^H = \exp(W \log_{\mathbf{o}}^K(\mathbf{x}^H)). \quad (6)$$

**Technical Challenge** However, the input from LLMs and the transformation results are in Euclidean space, we need to apply an additional exponential map and a logarithmic map on the basis of Equation (1) to align the Euclidean representation. This leads to the expression:

$$
\begin{aligned}
\mathbf{z}^E = W_{\text{LoRA}}(\mathbf{x}^E) &= W\mathbf{x}^E + \Delta W\mathbf{x}^E \\
&= W\mathbf{x}^E + \log_{\mathbf{o}}^K(\exp_{\mathbf{o}}^K(\underbrace{BA \overbrace{\log_{\mathbf{o}}^K(\exp_{\mathbf{o}}^K(\mathbf{x}^E))}^{\mathbf{x}^E}}_{\text{Transformation on } \mathbf{x}^E})) \\
&= W\mathbf{x}^E + BA\mathbf{x}^E,
\end{aligned}
\quad (7)
$$

which simplifies back to the original LoRA, rendering the method ineffective for our purposes.

**Direct Lorentz Low-rank Transformation (LLR)** To address this challenge, we perform low-rank adaptation directly on the hyperbolic manifold without utilizing tangent space:

$$
\begin{aligned}
\mathbf{z}^E = W_{\text{LoRA}}(\mathbf{x}^E) &= W\mathbf{x}^E + \Delta W\mathbf{x}^E \\
&= W\mathbf{x}^E + \log_{\mathbf{o}}^K(\underbrace{\mathbf{LLR}(BA, \overbrace{\exp_{\mathbf{o}}^K(\mathbf{x}^E)}^{\mathbf{x}^H})}_{\text{Transformation on } \mathbf{x}^H}),
\end{aligned}
\quad (8)
$$

where $\mathbf{LLR}$ represents the direct Lorentz Low-Rank Transformation which operate the hyperbolic representation $\mathbf{x}^H$ directly,

$$\mathbf{LLR}(BA, \mathbf{x}^H) = (\sqrt{\|B\mathbf{y}_*^H\|_2^2 + K}, B\mathbf{y}_*^H), \text{where } \mathbf{y}^H = (\sqrt{\|A\mathbf{x}_*^H\|_2^2 + K}, A\mathbf{x}_*^H), \quad (9)$$

We consider two transformations in our design, with $\mathbf{u}$ representing both $\mathbf{x}$ and $\mathbf{y}$: (1) $\mathbf{u}_*^H = \mathbf{u}_s$. (2) $\mathbf{u}_*^H = \mathbf{u}$. The first transformation only modifies the space-like dimension in special relativity, akin to a Lorentz rotation. The second transformation affects both time-like and space-like dimensions, similar to a Lorentz boost. In both cases, it can be verified that $\mathbf{LLR}(BA, \mathbf{x}^H) \in \mathcal{L}^n$. The linear

transformation is inspired by hyperbolic neural networks (Chen et al., 2021; Yang et al., 2024; Dai et al., 2021). For efficient integration with LLMs, the transformation removes normalization and non-linear activation term in (Chen et al., 2021), varying curvatures in (Yang et al., 2024), and orthogonal constraints in (Dai et al., 2021). Our main contribution lies in applying hyperbolic low-rank adaptation for LLMs, while the specific linear transformation itself is flexible—other transformations on the manifold could also be compatible with our approach.

In summary, our proposed method, HypLoRA, initially uses the exponential map to project the original Euclidean representation into hyperbolic space, applies a low-rank Lorentz transformation, and then employs the logarithmic map to revert to Euclidean space. By adapting in the hyperbolic domain, HypLoRA captures more complex hierarchical relationships than traditional Euclidean-based methods, as detailed in Proposition 5.1. Additionally, the low-rank nature of the adaptation matrices $A$ and $B$ promotes parameter efficiency, making HypLoRA well-suited for LLMs.

**Time Complexity** HypLoRA has similar theoretical time complexity as the Euclidean LoRA, which is $\mathcal{O}(r \cdot (d + k))$, where $d$ and $k$ represent the input and output dimensions, respectively. However, in practical implementation, HypLoRA introduces additional computations due to the logarithmic and exponential maps. These additional operations, nevertheless, can be completed within $\mathcal{O}(N)$ where the N is the number of input tokens.

**Proposition 5.1.** *Let* $\mathbf{x}$ *represent the input token embeddings, with* $\|\mathbf{x}\|$ *denoting their norms. HypLoRA modifies the query and key updates by introducing higher-order terms that depend on* $\|\mathbf{x}\|$. *This dependence enables HypLoRA to capture the hierarchical relationships in token embeddings. As a result, HypLoRA aligns with the intrinsic geometry of token embeddings.*

### 5.1 EXPERIMENTAL SETTINGS

**Dataset** The experimental setup closely follows the methodology in (Hu et al., 2023). The fine-tuning training set is composed of data from GSM8K (Cobbe et al., 2021), MAWPS, MAWPS-single (Koncel-Kedziorski et al., 2016), and 1,000 examples from AQuA (Ling et al., 2017). To further enhance reasoning capabilities, step-by-step rationales generated by ChatGPT are incorporated into the training samples, as done in (Hu et al., 2023). This results in a dataset of 10K math reasoning samples, named Math-10K, for training purposes. The test datasets include GSM8K (Cobbe et al., 2021), AQuA (Ling et al., 2017), MAWPS (Koncel-Kedziorski et al., 2016), and SVAMP (Patel et al., 2021). While the same datasets are used for training, there is no overlap between the training and test sets.

**Model Comparison** We include the LLaMA-7B and LLaMA-13B base models, as discussed in (Hu et al., 2023), along with the recently released Gemma-7B and LLaMA3-8B models, which are fine-tuned using LoRA for comparison. For fine-tuning methods, we evaluate several techniques, including Prefix-Tuning (Li & Liang, 2021), Series Adapter (Houlsby et al., 2019), LoRA (Hu et al., 2021), and Parallel Adapter (He et al., 2021). Additionally, we compare with DoRA (Liu et al., 2024), a recent competitive method.

**Implementation Details** The exponential map transforms the original input space using an exponential operator, as also noted in (Desai et al., 2023). To prevent numerical overflow, we first apply L2 normalization to the input before using the exponential map in Equation (8), then rescale it with a learnable norm scaling factor. The curvature for our proposed HypLoRA is treated as a hyperparameter, with values searched from the set {0.1, 0.5, 1.0, 2.0}. Following the procedure in (Chami et al., 2019), to correctly use the exponential map, we append a zero to the beginning of the input vector $\mathbf{x}$, forming $\mathbf{x}^E$. After applying the logarithmic map, the output vector $\mathbf{z}$ will have an additional dimension with a zero value. To maintain consistency with the original input space, we remove this extra dimension from $\mathbf{z}$. It is important to note that the final results are micro-averaged across datasets, which contain varying numbers of questions, such as 1,319 in GSM8K and 238 in MAWPS. In micro-averaging, each prompt is treated equally. For the LoRA implementation, we insert it into both the Multi-head Attention and MLP layers of the base model. All experiments are conducted on a single GPU, using either the A40 (40G) or A100 (80G).

### 5.2 EXPERIMENTAL RESULTS

Table 3 summarizes our key experimental outcomes, highlighting both the baseline model performance and the improvements from incorporating adapters. Since our experimental setup and dataset selection

Table 3: Accuracy comparison of various LLMs using PEFT methods on arithmetic reasoning tasks. Results marked with an asterisk (*) are sourced from Hu et al. (Hu et al., 2023).($^\dagger$) denotes our reproduced results on LoRA. The LoRA results for LLaMA3-8B and Gemma-7B are derived using the hyperparameters specified in the same study. The percentage following each dataset indicates the proportion of prompts relative to the total number of inference prompts. M.AVG represents the micro-average accuracy. OOT denotes evaluations exceeding 24 hours on an A100 GPU, while None refers to the base model without any PEFT method applied. NA stands for Not Applicable.

| Model | PEFT Method | MAWPS(8.5%) | SVAMP(35.6%) | GSM8K(46.9%) | AQuA(9.0%) | M.AVG |
|---|---|---|---|---|---|---|
| GPT-3.5 | None | **87.4** | **69.9** | **56.4** | **38.9** | **62.3** |
| LLaMA-7B | None | 51.7 | 32.4 | 15.7 | 16.9 | 24.8 |
| | Prefix* | 63.4 | 38.1 | 24.4 | 14.2 | 31.7 |
| | Series* | 77.7 | 52.3 | 33.3 | 15.0 | 42.2 |
| | Parallel* | 82.4 | 49.6 | 35.3 | 18.1 | 42.8 |
| | LoRA* | 79.0 | 52.1 | 37.5 | 18.9 | 44.6 |
| | LoRA$^\dagger$ | 81.9 | 48.2 | 38.3 | 18.5 | 43.7 |
| | DoRA | 80.0 | 48.8 | 39.0 | 16.4 | 43.9 |
| | **HypLoRA (Ours)** | 79.0 | 49.1 | 39.1 | 20.5 | 44.4 |
| LLaMA-13B | None | 65.5 | 37.5 | 32.4 | 15.0 | 35.5 |
| | Prefix* | 66.8 | 41.4 | 31.1 | 15.7 | 36.4 |
| | Series* | 78.6 | 50.8 | 44.0 | 22.0 | 47.4 |
| | Parallel* | 81.1 | 55.7 | 43.3 | 20.5 | 48.9 |
| | LoRA* | 83.6 | 54.6 | 47.5 | 18.5 | 50.5 |
| | LoRA$^\dagger$ | 83.5 | 54.7 | 48.5 | 18.5 | 51.0 |
| | DoRA | 83.0 | 54.6 | OOT | 18.9 | NA |
| | **HypLoRA (Ours)** | 83.2 | 54.8 | 49.0 | 21.5 | 51.5 |
| Gemma-7B | None | 76.5 | 60.4 | 38.4 | 25.2 | 48.3 |
| | LoRA | 91.6 | 76.2 | 66.3 | 28.9 | 68.6 |
| | DoRA | 91.7 | 75.9 | 65.4 | 27.7 | 68.0 |
| | HypLoRA (Ours) | 91.5 | 78.7 | 69.5 | 32.7 | 71.3 |
| LLaMA3-8B | None | 79.8 | 50.0 | 54.7 | 21.0 | 52.1 |
| | LoRA | **92.7** | 78.9 | 70.8 | 30.4 | 71.9 |
| | DoRA | 92.4 | 79.3 | 71.3 | 33.1 | 72.5 |
| | **HypLoRA (Ours)** | 91.6 | **80.5** | **74.0** | **34.2** | **74.2** |

closely align with those used by Hu et al. (Hu et al., 2023), we reference their results directly. For the new base models, like Gemma-7B and LLaMA3-8B, we maintained the same training strategy for consistency. Each experiment was run three times, and we reported the average results. We have the following findings:

**Overall Performance of HypLoRA and Challenging Datasets** The accuracy from GPT-3.5 indicates that GSM8K and AQuA are among the more challenging datasets in this evaluation. The performance of the models and the fine-tuned results is strongly related to the difficulty level of the datasets. It is as complex problems require more complex reasoning and a better understanding of the underlying structure of the problem. Nonetheless, HypLoRA consistently outperforms the baseline methods across various base models. Notably, the overall performance improvement reaches up to 2.3% on LLaMA3-8B and 3.9% on Gemma-7B against the best competitors. In addition, HypLoRA method excels on these complex datasets, AQuA and GSM8K, where it achieves an improvement of up to 13.0% on the AQuA dataset and 4.8% on the GSM8K dataset with the Gemma-7B model. This significant gain reflects the advantage of introducing hyperbolic geometry, as its inherent geometric properties make it better suited for capturing complex, hierarchical structured data. The analysis in Appendix D demonstrates that HypLoRA implicitly introduces high-order interaction terms and highlights higher-order terms proportional to the token norm, correlating with more specific tokens in the token hierarchy. This enables the model to focus on more tokens and better comprehend complex relationships. Consequently, HypLoRA effectively leverages the hierarchical and hyperbolic structure of the data, resulting in improved performance on challenging reasoning tasks.

**Performance with DoRA and on MAWPS Dataset** The computational complexity of DoRA leads to timeouts in evaluations, such as for GSM8K on LLaMA-13B. Despite this, HypLoRA consistently performs as well or better than DoRA in completed evaluations, offering comparable results with much lower computational overhead. On the MAWPS dataset, the performance improvements of HypLoRA are less pronounced compared to other datasets, and in some instances, it falls below the baseline results. This may be attributed to using the same curvature during fine-tuning, which might not be suitable for all datasets. To address this, future work will focus on prompt-adaptive curvature

techniques. Despite this limitation, HypLoRA has demonstrated significant improvements over the base model across the majority of datasets evaluated.

## 5.3 Ablation Study and Parameter Analysis

**Ablation Study** We use the tangent-space method described in Equation (7) as a basis for conducting an ablation study. The primary difference between the tangent-space method and the proposed HypLoRA lies in the approach used for the Low-rank Transformation. Through this compar-

Table 4: Ablation Study

| Model | Methods | MAWPS | SVAMP | GSM8K | AQuA | M.AVG |
|-------|---------|-------|-------|-------|------|-------|
| **Gemma 7B** | Tangent | 91.2 | 77.9 | 67.6 | 30.5 | 69.9 |
| | HypLoRA (I) | 93.2 | 78.3 | 68.5 | 33.2 | 70.9 |
| | HypLoRA (II) | 91.5 | 78.7 | 69.5 | 32.7 | **71.3** |
| **LLaMA 8B** | Tangent | 91.2 | 79.2 | 72.3 | 30.3 | 72.6 |
| | HypLoRA (I) | 91.6 | 80.2 | 74.1 | 33.8 | 74.1 |
| | HypLoRA (II) | 91.6 | 80.5 | 74.0 | 34.2 | **74.2** |

ison, we can determine the effectiveness and benefits of the direct Lorentz Low-rank approach. Furthermore, compared to the Euclidean LoRA, both Equation (7) and the proposed HypLoRA incorporates an additional rescaling operation, as discussed in Section 5.1. Considering that the tangent-space method can be reduced to the general LoRA form, it can be viewed as an additional rescaling operation combined with the vanilla LoRA. By making these comparisons, we can evaluate the effectiveness of normal rescaling.

Table 4 presents our results. HypLoRA (I) denotes the first transformation (Lorentz rotation) on space-like dimensions and HypLoRA (II) denotes the second transformation (Lorentz boost) on the whole dimensions. We observe that the tangent space method shows improvement over the original LoRA, which is expected since the rescaling step introduces more flexibility. Due to the exponential effects in hyperbolic geometry, this rescaling step is necessary. Comparing the results of HypLoRA with the tangent space method, we can see the significant impact of introducing hyperbolic geometry, with the main improvements attributed to this incorporation. We also observe that these two transformations obtain similar performances.

**The Impact of Curvature on Performance** Curvature in hyperbolic space is a key hyperparameter in HypLoRA, directly affecting its capacity to model underlying structures and geometries. To evaluate its impact, we experimented with different curvature values on the Gemma 7B model, as shown in Table 5, where curvature is defined as $-1/K$. Our results demonstrate that overall model performance remains relatively stable across various curvature settings. Notably, determining the optimal curvature value was straightforward,

Table 5: Results for varying of K

| K | 0.1 | 0.5 | 1.0 | 2.0 |
|---|-----|-----|-----|-----|
| MAWPS | 89.8 | 91.7 | 91.5 | 90.8 |
| SVAMP | 78.1 | 77.3 | 78.7 | 78.6 |
| GSM8K | 68.5 | 67.5 | 69.5 | 68.5 |
| AQuA | 31.9 | 34.3 | 32.7 | 31.1 |

with a final value of 1.0 proving to be the best. In future work, we will consider exploring the data-informed curvature method to make the fine-tuning more adaptive.

**Inference Efficiency** In Section 5, we analyze the time complexity of our approach, which remains consistent with that of LoRA. However, during actual inference, HypLoRA incurs additional computational overhead due to operations such as the exponential and logarithmic mappings. These operations introduce some additional runtime, particularly for larger models. The GPU hours for inference on four datasets are presented in Figure 3. Despite this overhead, our method demonstrates improved efficiency when compared to the previous competitive model, DoRA. Notably, HypLoRA still outperforms DoRA in terms of both runtime and overall efficiency. In future work, we plan to investigate more efficient approaches to further reduce this computational cost, with the goal of minimizing the impact of these transformations on overall inference time.

**Case Study** The introduction of hyperbolic space allows the model to capture more token information and accurately comprehend the semantics of the prompt, leading to more effective solutions for complex reasoning tasks. Table 6 provides examples to demonstrate HypLoRA's capabilities. In this case, LoRA fails to grasp the true nature of the computational task, while HypLoRA correctly interprets the problem by recognizing the three-week timeframe and appropriately distributing the

required time. This highlights the superior ability of semantics understanding and the ability to handle multi-step reasoning.

Furthermore, the introduction of hyperbolic space enhances the model's ability to comprehend hierarchical relationships between tokens, providing better separation of long-tail tokens for prediction. This is due to the exponential growth properties of hyperbolic space, which fit these tokens more effectively and yield more accurate reasoning outcomes. Additional case studies are provided in Appendix E.

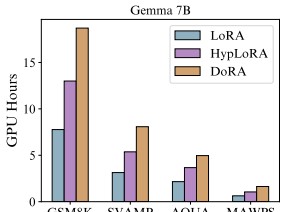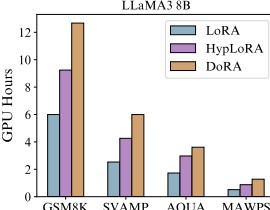

Figure 3: Illustration of GPU (A100) hours during inference

Table 6: Example problem with solutions from LoRA and HypLoRA

| | |
|---|---|
| **Question** | It takes 240 minutes of walking to break in a new pair of shoes. Jason wants to try out for the track team in three weeks. If he can walk 4 days a week to break in the new shoes, how long does he have to spend walking each day? **Answer:** 20.0 minutes |
| **LoRA** | To find out how long Jason has to spend walking each day, we need to divide the total number of minutes it takes to break in the shoes by the number of days he has to do it. Jason has 240 minutes divided by 4 days per week. This results in 60 minutes per day. **Answer: 60**. |
| **HypLoRA** | First, we need to find how many minutes Jason needs to walk in total to break in the new shoes: 240 minutes. Next, we find how many minutes Jason needs to walk each week over three weeks: 240 minutes divided by 3 weeks equals 80 minutes per week. Then, to find how many minutes he needs to walk each day: 80 minutes per week divided by 4 days equals 20 minutes per day. **Answer: 20**. |

## 6 CONCLUSION

In this study, we investigated the non-Euclidean properties of token embeddings in LLMs. Our analysis token embedding has strong hyperbolic characteristics. Building on these findings, we proposed HypLoRA, a hyperbolic low-rank adaptation method that incorporates hyperbolic geometry into the fine-tuning process of LLMs. Extensive experiments showed that HypLoRA significantly improves LLM performance on arithmetic reasoning tasks, particularly on complex datasets. By leveraging the hyperbolic structure of the data, HypLoRA enhances the model's ability to capture and utilize intricate relationships, leading to better reasoning capabilities.

**Limitation and Future Work** In this study, we employed a consistent curvature across all prompts during fine-tuning, which simplified the implementation and enhanced efficiency. However, this uniform approach may not be optimal when applied to different datasets simultaneously. Our future work will explore more adaptive fine-tuning techniques that can better accommodate the unique characteristics of different prompts. Additionally, due to the computational overhead introduced by the exponential and logarithmic maps, this is inevitable when transitioning from the original Euclidean space to a hyperbolic space. We will explore more efficient methods to reduce this computational cost in future work.

Despite these challenges, our research provides a thorough examination of token embedding distributions from a non-Euclidean perspective and offers valuable insights. The fine-tuning method we proposed holds significant potential for advancing geometrically inspired models, contributing to the ongoing development of more effective LLMs.

**Reproducibility Statement** Due to the inherent randomness in large model inference, answers on individual prompts may slightly different, but overall results are consistent. We averaged results over multiple runs. For LoRA, we rerun LoRA on LLaMA-7B/13B, though performance on SVAMP with LLaMA-7B was lower than reported. All DoRA experiments were run on A100 GPUs due to long inference times on A40, with LLaMA-13B still exceeding 24 hours.

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

## A  INVESTIGATION ON MORE DATASETS

In the main text, we focused on token distribution for GSM8K datasets. Here, we provide more token distribution for the AQUA and MAWPS mathematical reasoning datasets. Besides, we extend this analysis to include common sense reasoning datasets, specifically OpenBookQA and WinoGrande. These results are shown in Figure 4. The findings align with the conclusions drawn in the main text.

## B  HYPERBOLICITY ON DIFFERENT METRIC SPACES

Table 2 presents the hyperbolicity values in both continuous (i.e., Sphere Space) and discrete metric spaces (i.e., Tree Graph, Scale-free Graph and Random Graph). We employ a consistent processing method, akin to the one mentioned in Section (4) for embedding spaces. Specifically, we sample 1000 4-tuples, compute the delta value for each, and then take the maximum value.

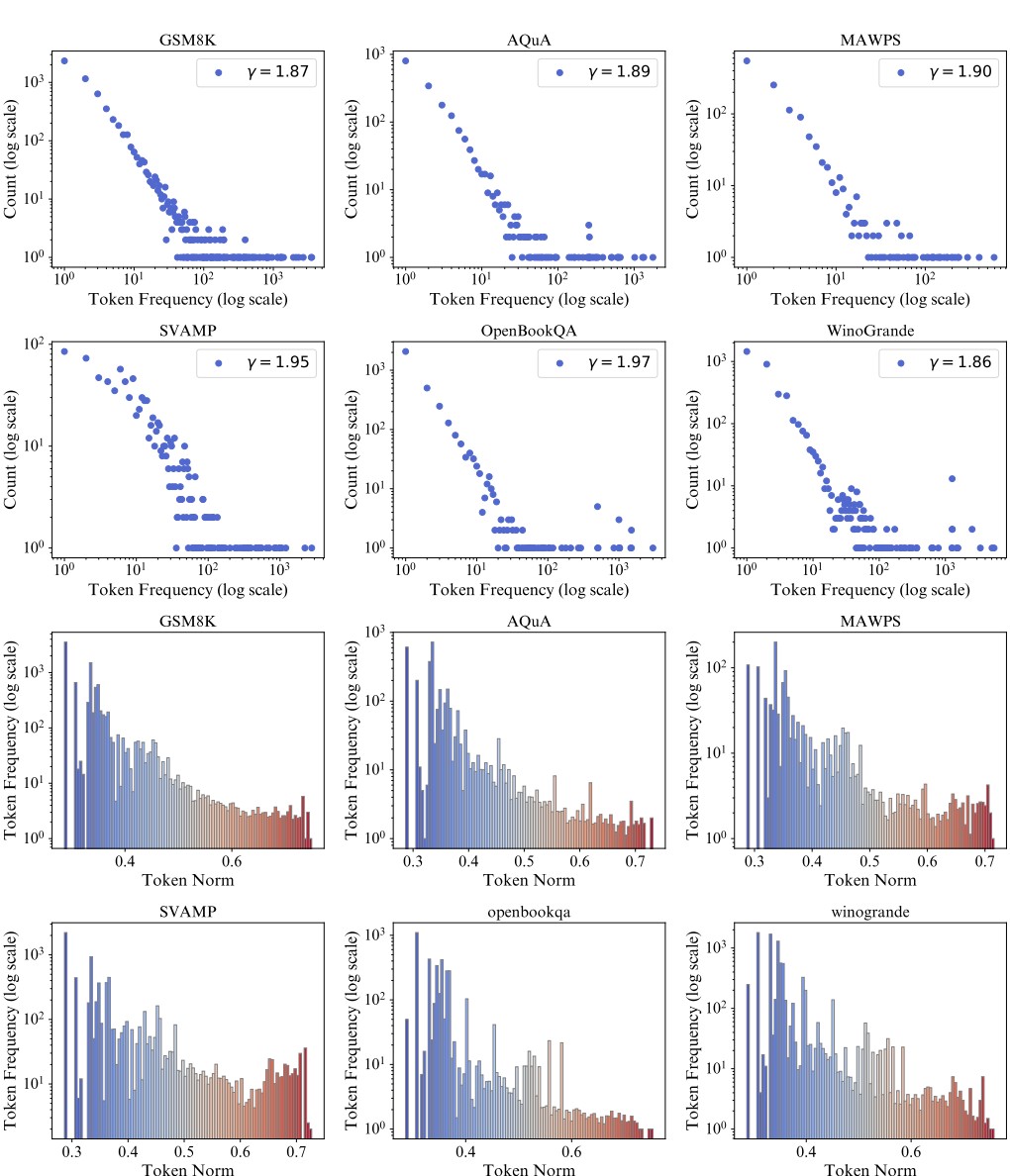

Figure 4: Token frequency distribution (top two rows) and token frequency vs. norm (bottom two rows) across different datasets in LLaMA3. The top two row show the power-law distribution of token frequencies with the decay rate ($\gamma$) annotated for each dataset. The bottom two rows illustrates the relationship between token frequency and token norm, binned and colored by frequency, where higher token norms correspond to lower frequencies.

For the sphere space, we use a two-dimensional model and calculate hyperbolicity based on their respective geodesic distances. The PubMed graph is sourced from Sen et al. (Sen et al., 2008). The tree graph and dense graph are generated using NetworkX (Hagberg et al., 2008). For these graphs, we first remove isolated nodes before performing our calculations in a consistent manner. The shortest path distance on the graph is used as the distance measure, analogous to the concept of geodesics in continuous spaces.

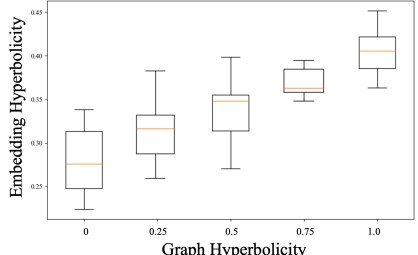

Figure 5: Correlation between graph and embedding hyperbolicity.

In this study, we utilize the Euclidean distance to compute the hyperbolicity of token embeddings, following the approach proposed by Khrulkov et al. (2020). To further validate the correctness of this method, we embed graphs with varying degrees of hyperbolicity into Euclidean space using a graph neural network (GNN) model and compute hyperbolicity based on the distances between embeddings. The results, presented in Figure 5, indicate a positive correlation between the hyperbolicity of the original graphs and that of the embeddings, although the values do not exactly coincide. Building on this observed relationship, we calculate the hyperbolicity of token embeddings as a proxy for estimating their underlying geometric structure. In this context, lower hyperbolicity values suggest a more tree-like geometric configuration.

## C EXPONENTIAL AND LOGARITHMIC MAP

The exponential and logarithmic maps serve as essential tools for projection between the local tangent space and the hyperbolic space. Consider a point $\mathbf{x} \in \mathcal{L}_K^n$ and a tangent vector $\mathbf{u} \in \mathcal{T}_\mathbf{x}\mathcal{L}_K^n$. The exponential map, denoted as $\exp_\mathbf{x}^K : \mathcal{T}_\mathbf{x}\mathcal{L}_K^n \to \mathcal{L}_K^n$, assigns to $\mathbf{u}$ the point $\exp_\mathbf{x}^K(\mathbf{u}) := \gamma(1)$, where $\gamma$ represents the unique geodesic that satisfies the initial conditions $\gamma(0) = \mathbf{x}$ and $\dot{\gamma}(0) = \mathbf{u}$. The exponential map can be explicitly expressed as follows:

$$\exp_\mathbf{x}^K(\mathbf{u}) = \cosh\left(\frac{\|\mathbf{u}\|_\mathcal{L}}{\sqrt{K}}\right)\mathbf{x} + \sqrt{K}\sinh\left(\frac{\|\mathbf{u}\|_\mathcal{L}}{\sqrt{K}}\right)\frac{\mathbf{u}}{\|\mathbf{u}\|_\mathcal{L}}, \tag{10}$$

where $\cosh$ and $\sinh$ represent the hyperbolic cosine and sine functions, respectively, and $\|\mathbf{u}\|_\mathcal{L}$ denotes the norm of the tangent vector $\mathbf{u}$ in the tangent space.

The logarithmic map $\log_\mathbf{u}^K(\mathbf{x}) : \mathcal{L}_K^n \to \mathcal{T}_\mathbf{u}\mathcal{L}_K^n$ plays an inverse role. It is defined by the equation:

$$\log_\mathbf{u}^K(\mathbf{x}) = \frac{\cosh^{-1}\left(-\frac{1}{K}\langle\mathbf{u}, \mathbf{x}\rangle_\mathcal{L}\right)}{\sinh\left(\cosh^{-1}\left(-\frac{1}{K}\langle\mathbf{u}, \mathbf{x}\rangle_\mathcal{L}\right)\right)}\left(\mathbf{x} + \frac{1}{K}\langle\mathbf{u}, \mathbf{x}\rangle_\mathcal{L}\mathbf{u}\right). \tag{11}$$

The exponential and logarithmic maps establish a bijective projection between the tangent space and hyperbolic space. Notably, $\log_\mathbf{x}^K(\exp_\mathbf{x}^K(\mathbf{u})) = \mathbf{u}$ and $\exp_\mathbf{u}^K(\log_\mathbf{u}^K(\mathbf{x})) = \mathbf{x}$. Consequently, Equation (7) will cancel out the hyperbolic operations. In addition, these operations are typically defined locally. However, in the context of hyperbolic representation and deep learning, for efficient computation, existing works usually use the origin point $\mathbf{o} := \{\sqrt{K}, 0, \dots, 0\} \in \mathcal{L}_K^n$ as a common reference point.

## D TRANSFORMATION ANALYSIS

*Proof.* Let $\mathbf{x} \in \mathbb{R}^n$ be an input token embedding, and $A, B \in \mathbb{R}^{n \times r}$ be low-rank matrices with rank $r \ll n$. Consider the hyperbolic space $\mathbb{H}^n$ with curvature $C = -\frac{1}{R^2}$, where $R > 0$ is the radius of curvature.

**(1) Exponential Map at the Origin (o):**

For a tangent vector $\mathbf{v} \in T_\mathbf{o}\mathbb{H}^n$:

$$\exp_\mathbf{o}(\mathbf{v}) = \left(\cosh\left(\frac{\|\mathbf{v}\|}{R}\right), \sinh\left(\frac{\|\mathbf{v}\|}{R}\right)\frac{\mathbf{v}}{\|\mathbf{v}\|}\right) \tag{12}$$

**(2) Logarithmic Map at the Origin ($\mathbf{o}$):**

For $\mathbf{x}^H = (x_0, \mathbf{x}_{\text{space}}) \in \mathbb{H}^n$:

$$\log_{\mathbf{o}}(\mathbf{x}^H) = R \cdot \operatorname{arcosh}\left(\frac{x_0}{R}\right) \frac{\mathbf{x}_{\text{space}}}{\sqrt{x_0^2 - R^2}} \tag{13}$$

**(3) Applying Low-Rank Transformations**: For simplicity, we analysis about the transformation on space-like coordinates *First Transformation*:

$$\begin{aligned} \mathbf{y}_{\text{space}}^H &= A^\top \mathbf{x}_{\text{space}}^H, \\ y_0^H &= \sqrt{R^2 + \left\| \mathbf{y}_{\text{space}}^H \right\|^2}. \end{aligned} \tag{14}$$

*Second Transformation*:

$$\begin{aligned} \mathbf{z}_{\text{space}}^H &= B^\top \mathbf{y}_{\text{space}}^H, \\ z_0^H &= \sqrt{R^2 + \left\| \mathbf{z}_{\text{space}}^H \right\|^2}. \end{aligned} \tag{15}$$

**(4) Mapping Back to Euclidean Space**:

The update to the query vector is:

$$\Delta Q^{\text{Hyp}} = R \cdot \operatorname{arcosh}\left(\frac{z_0^H}{R}\right) \frac{\mathbf{z}_{\text{space}}^H}{\left\| \mathbf{z}_{\text{space}}^H \right\|} \tag{16}$$

**(5) Approximations Incorporating Token Norms**:

From the investigation, we know that token norms $\|\mathbf{x}\|$ are correlated with their specificity in the hierarchical structure: tokens with larger norms represent more specific concepts.

For small $\frac{\|\mathbf{x}\|}{R}$, we use the Taylor series expansions:

- $\cosh\left(\frac{z}{R}\right) \approx 1 + \frac{z^2}{2R^2}$.

- $\sinh\left(\frac{z}{R}\right) \approx \frac{z}{R} + \frac{z^3}{6R^3}$.

Therefore, the spatial component after the exponential map is:

$$\mathbf{x}_{\text{space}}^H \approx \frac{\mathbf{x}}{R} + \frac{\|\mathbf{x}\|^2}{6R^3}\mathbf{x} \tag{17}$$

**(6) Applying the Transformations**:

*First Transformation*:

$$\mathbf{y}_{\text{space}}^H \approx \frac{A^\top \mathbf{x}}{R} + \frac{\|\mathbf{x}\|^2}{6R^3}A^\top \mathbf{x} \tag{18}$$

*Second Transformation*:

$$\mathbf{z}_{\text{space}}^H \approx \frac{(BA)\mathbf{x}}{R} + \frac{\|\mathbf{x}\|^2}{6R^3}(BA)\mathbf{x} \tag{19}$$

**(7) Approximating the Logarithmic Map**:

*Compute $z_0^H$:*

$$z_0^H = \sqrt{R^2 + \left\| \mathbf{z}_{\text{space}}^H \right\|^2} \approx R + \frac{\left\| \mathbf{z}_{\text{space}}^H \right\|^2}{2R} \tag{20}$$

*Compute* $\text{arcosh}\left(\dfrac{z_0^H}{R}\right)$:

For small $\delta = \dfrac{\left\| \mathbf{z}_{\text{space}}^H \right\|^2}{2R^2}$:

$$\text{arcosh}\left(1 + \delta\right) \approx \sqrt{2\delta} = \frac{\left\| \mathbf{z}_{\text{space}}^H \right\|}{R} \tag{21}$$

*Final Expression for $\Delta Q^{\text{Hyp}}$:*

$$\Delta Q^{\text{Hyp}} \approx \mathbf{z}_{\text{space}}^H. \tag{22}$$

**(8) Comparing HypLoRA and LoRA Updates**:

*HypLoRA Update*:

$$\Delta Q^{\text{Hyp}} \approx \frac{(BA)\mathbf{x}}{R} + \frac{\|\mathbf{x}\|^2}{6R^3}(BA)\mathbf{x}. \tag{23}$$

*LoRA Update*:

$$\Delta Q^{\text{LoRA}} = \frac{(BA)\mathbf{x}}{R}. \tag{24}$$

*Difference*:

$$\Delta Q^{\text{Hyp}} - \Delta Q^{\text{LoRA}} = \frac{\|\mathbf{x}\|^2}{6R^3}(BA)\mathbf{x}. \tag{25}$$

**(9) Impact of Token Norms on Higher-Order Terms**:

Since $\|\mathbf{x}\|$ reflects the specificity of the token in the hierarchical structure (larger norms correspond to more specific tokens), the higher-order term $\dfrac{\|\mathbf{x}\|^2}{6R^3}(BA)\mathbf{x}$ becomes significant for tokens representing specific concepts.

**(10) Impact on Attention Scores**:

The HypLoRA attention scores are computed as:

$$\text{Scores}_{\text{HypLoRA}} = \frac{\left(Q^{\text{orig}} + \Delta Q^{\text{Hyp}}\right)\left(K^{\text{orig}} + \Delta K^{\text{Hyp}}\right)^\top}{\sqrt{d_k}}. \tag{26}$$

where $\Delta K^{\text{Hyp}}$ is derived similarly.

The difference in attention scores includes higher-order terms dependent on $\|\mathbf{x}\|^2$:

$$\Delta\,\text{Scores} = \text{Scores}_{\text{HypLoRA}} - \text{Scores}_{\text{LoRA}}. \tag{27}$$

These higher-order terms allow HypLoRA to capture more complex, hierarchical relationships, particularly for tokens with larger norms (more specific tokens).

$\square$

**Remark D.1.** *Alignment with Token Hierarchy: The higher-order terms in HypLoRA's updates are proportional to $\|\mathbf{x}\|^2$, which, according to our investigation, correlates with the specificity of tokens in the hierarchical structure. This means HypLoRA places greater emphasis on more specific tokens, enhancing its ability to model detailed relationships.*

**Role of Curvature** *$C$: The curvature scales the higher-order corrections, and in spaces with larger negative curvature (smaller $R$), these terms become more pronounced, aligning with the hyperbolic nature of token embeddings.*

**Effectiveness of HypLoRA**: *By incorporating these higher-order terms, HypLoRA effectively leverages the inherent hierarchical and hyperbolic structure of token embeddings, leading to improved performance, especially the problems that require complex reasoning. This is also the reason why the proposed method has a better performance on more challenging datasets.*

## E  CASE STUDY

In this section, we present examples to demonstrate the advantages of incorporating HypLoRA. As the examples show in Table 7 and 8, HypLoRA consistently provides more accurate reasoning compared to LoRA. For instance, in Example 2, LoRA incorrectly calculates the sum of the test scores as 380 instead of the correct total of 400, which leads to an incorrect average. This mistake highlights a limitation of large models like LoRA, which do not perform arithmetic operations as precisely as computers but rather approximate probability distributions. While the values 380 and 400 may seem close in Euclidean space, the hyperbolic geometry utilized by HypLoRA more effectively handles such variations, particularly in "tail" distributions, allowing for greater separation between close values and thus leading to more accurate reasoning. A similar issue arises in Example 3, where LoRA incorrectly computes $11,600 - 9,800$, resulting in a wrong conclusion about the load capacity of Gary's truck. HypLoRA, on the other hand, correctly performs the calculation and arrives at the accurate answer. These examples demonstrate how the hyperbolic space used by HypLoRA enables better handling of complex reasoning tasks, improving both accuracy and reliability in such scenarios.

Table 7: Case Study 2

| | |
|---|---|
| **Question** | To have the car for the weekend, Wilson's report card needs to show that he received 80 or higher in his math class. His math grade is calculated by the average of 5 test scores. On those tests, he received: 65, 94, 81, 86, and 74. What is his current math grade? **Answer:** 80.0 |
| **LoRA** | To find Wilson's current math grade, we need to take the average of his 5 test scores. First, we add up all 5 test scores: $65 + 94 + 81 + 86 + 74 = 380$. Then, we divide the sum by the number of tests: $380 \div 5 = 76$. Therefore, Wilson's current math grade is 76. **Answer: 76**. |
| **HypLoRA** | To find Wilson's current math grade, we need to take the average of his 5 test scores. To do this, we add up the scores and divide by 5: $65 + 94 + 81 + 86 + 74 = 400$. Then, we divide by 5: $400 \div 5 = 80$. Therefore, Wilson's current math grade is 80. **Answer: 80**. |

Table 8: Case Study 3

| | |
|---|---|
| **Question** | Gissela, Gordy, and Gary are truck drivers. Gissela has a truck large enough to haul 4,000 pounds of gravel. Gordy's truck can haul 800 pounds more than Gissela's truck. When Gary brings his truck and joins Gissela and Gordy, the three trucks combined can haul a total of 11,600 pounds of gravel. How many pounds of gravel can Gary's truck carry? **Answer: 2,800 pounds** |
| **LoRA** | Let's start by assigning variables to represent the amount of gravel each truck can haul: Gissela's truck hauls 4,000 pounds. Gordy's truck hauls $4,000 + 800 = 4,800$ pounds. Now, we set up an equation for the total amount of gravel the three trucks can haul: $4,000 + 4,800 + x = 11,600$, where $x$ is the amount Gary's truck hauls. Simplifying the equation: $9,800 + x = 11,600$, we subtract $9,800$ from both sides to get $x = 11,600 - 9,800 = 1,800$. Therefore, Gary's truck hauls 1,800 pounds. **Answer: 1,800**. |
| **HypLoRA** | Gissela's truck can haul 4,000 pounds of gravel. Gordy's truck can haul 800 more pounds than Gissela's truck, so it can haul 4,800 pounds of gravel. Together, Gissela and Gordy's trucks can haul 8,800 pounds of gravel. If the three trucks combined can haul 11,600 pounds, then Gary's truck can haul $11,600 - 8,800 = 2,800$ pounds of gravel. **Answer: 2,800**. |

