# OpenReview forum: "Hyperbolic Fine-tuning for Large Language Models"
_ICLR.cc/2025/Conference — Submitted to ICLR 2025_

### Official Review · Reviewer_yGwL · 2024-10-18

**Soundness:** 3
**Presentation:** 4
**Contribution:** 3
**Rating:** 6
**Confidence:** 3

**Summary:**

This paper explores the non-Euclidean properties of LLMs, demonstrating that token frequency adheres to a power-law distribution and that token embeddings reveal a significant hyperbolic structure, suggesting an underlying tree-like arrangement in the embedding space. It introduces a novel approach, Hyperbolic Low-Rank Efficient Fine-Tuning (HypLoRA), designed to fine-tune LLMs effectively in hyperbolic space while circumventing the cancellation effects associated with exponential and logarithmic maps. HypLoRA markedly improves LLM performance on reasoning tasks, especially for complex problems.

**Strengths:**

1. This paper claims to be the first study on fine-tuning LLMs in hyperbolic space. It investigates the token embedding properties of LLMs and proposes the HypLoRA method based on the analysis. Overall, the work makes a significant contribution.
2. This paper is well-organized and well-written, encompassing analysis, methodology, experiments, and conclusions, with a clear overall presentation and fluent language.
3. The content of this paper is rich and comprehensive, with thorough comparative experiments and analyses.

**Weaknesses:**

1. In Introduction and Related Work, the authors claimed existing works have not attempted to study LLM embeddings in the context of non-Euclidean geometry. However, there are studies on hyperbolic space learning of language models like Chen W, et al, 2024 (https://ieeexplore.ieee.org/document/10542420). It may not directly relate to LLMs, but I still believe that these methods should be considered as relevant work and comparative approaches, as the main idea of your study is on hyperbolic learning. An actionable suggestion is trying to discuss how your approach differs from or builds upon these existing methods for smaller language models, and explain why their technique is specifically suited for LLMs.
2. In Introduction part (Line 95-97), a brief explanation would be better about how the proposed HypLoRA is designed and what features of HypLoRA leads to its consideration of token hierarchies and minimizing computational costs.
3. In Section 4.1, how you obtain the distribution of token frequencies? Though the computation may be simple, a brief illustration would still be better to provide a clearer presentation.
4. In Section 4.2 (Line 282-283), can the power-law frequency distribution and significant hyperbolicity completely indicate a tree-like hierarchical structure of token embeddings? Are there any previous studies on the properties of tree-like hierarchical structures can prove this conclusion? A more rigorous analysis would be better to give the conclusion of the investigation. I think the authors could either provide references to studies that support this connection between power-law distributions, hyperbolicity, and tree-like structures, or to clarify that this is a hypothesis that requires further investigation.
5. Table 3 should provide more direct comparison between HypLoRA and other PEFT methods, expecially LoRA, by using highlighting marks for easier understanding.
6. In the inference efficiency part of Section 5.3, it would be better to record both inference and fine-tuning efficiency for a more comprehensive analysis.

**Questions:**

1. In Section 5, why directly applying low-rank adaptation on hyperbolic manifold works? What is the main purpose of using tangent space from previous work? Is LLR already an existing and universal strategy of reducing costs of learning hyperbolic representation?
2. In Table 3, from my view, HypLoRA seems not have obvious performance advantages than other methods on all datasets? Maybe the performance of hyperbolic fine-tuning is task-dependent? So I would like to see more performance comparison on other tasks rather than just arithmetic reasoning.
3. In case study part of Section 5.3, as you mentioned the hyperbolic learning is good at capturing tree-like nature of language, would it be better to provide a specific example that demonstrates how HypLoRA better captures hierarchical relationships in language compared to baseline methods? Such as showing a case where HypLoRA correctly handles a complex hierarchical concept that other methods struggle with.

---

> ### Author Response · Authors · 2024-11-19
> **Response (1/3)**
>
> We appreciate your valuable comments and suggestions. Below, we will represent the question/concerns with "**C:**" and our responses with "**R:**".
>
> ---
>
> **C**: Discussion with existing Chen's work [1]
>
> **R**: We appreciate your careful attention to related work, particularly regarding Chen et al.'s work. We will discuss the following in our revised version. Let us clarify the key differences and our contributions:
>
> **Motivation**. The main motivation for Chen's work is based on that PLMs’ intermediate representations are more effectively probed when represented by hyperbolic space, which was from a previous work [2]. While our approach is grounded in empirical and quantitive analysis (including token frequencies, norm, and hyperbolicity) of token embedding characteristics.
>
> **Objective**. Chen et al. address the challenge of pre-training BERT-scale models in hyperbolic space, whereas our work focuses on efficiently adapting large-scale models (7B-13B parameters) through hyperbolic fine-tuning while preserving their pre-trained capabilities.
>
> **Techniques**. Chen's work is to build a hyperbolic Bert while our method focuses on Hyperbolic Low rank adaptation. The overlap part is the Lorentz transformation. Chen's Lorentz transformation is given by
> $$
> \mathbf{W} \otimes^K \mathbf{x}:=\left(\sqrt{|\phi(\mathbf{x}, \mathbf{W}, \mathbf{v})|^2+K} ; \phi(\mathbf{x}, \mathbf{W}, \mathbf{v})\right), \phi(\mathbf{x}, \mathbf{W}, \mathbf{v})=\frac{\alpha}{|\mathbf{W h}(\mathbf{x})|} \mathbf{W} h(\mathbf{x}), \alpha=\sqrt{\left(\lambda \sigma\left(\mathbf{v}^T \mathbf{x}+b\right)+1+\epsilon\right)^2-1}
> $$
> where $\mathbf{W}, \mathbf{v}, b, \lambda$ are learnable parameters, $\sigma(\cdot)$ is the sigmoid function,  $h(\cdot)$ is the composition of dropout and activation function.
>
> In contrast, our HypLoRA employs a more straightforward transformation,
> $$
> \mathbf{W} \otimes^K \mathbf{x}=(\sqrt{\|{W\mathbf{x}}^H\|^2_2 + K}, {W\mathbf{x}}^H),
> $$
> This approach eliminates additional learnable parameters, normalization, and activation functions, making it particularly suitable for efficient LLM adaptation while maintaining the benefits of hyperbolic geometry.  Additionally, we tried Chen's transformation method and different lambda parameters, but it consistently resulted in a NAN issue.
>
> [1]Chen, Weize, et al. "Hyperbolic Pre-Trained Language Model." *IEEE/ACM Transactions on Audio, Speech, and Language Processing* (2024).
>
> [2]Chen, Boli, et al. "Probing BERT in hyperbolic spaces." arXiv preprint arXiv:2104.03869 (2021).
>
> ---
>
> **C**: In the introduction subsection, add brief explanation of the HypLoRA's design,  token hierarchies, and computational cost reduction.
>
> **R**: Thank you for this valuable suggestion to enhance the introduction. We will add this in our revised version.
>
> ---
>
> **C**: How to obtain the token frequencies?
>
> **R**: Thank you for raising this point. The token frequency distribution was computed through the following process:
>
> 1. We first tokenized all prompts in each dataset (e.g., GSM8K) using the respective model's tokenizer
> 2. For each unique token, we counted its total occurrences across all prompts in the dataset
> 3. We then sorted these frequency counts in descending order and plotted them on a log-log scale
> 4. The power-law exponent (γ ≈ 1.9) was estimated using the Powerlaw Package (Alstott et al., 2014), which fits the frequency distribution to p(x) ∝ x^(-γ).
>
> We will add this explanation to Section 4.1 to make the methodology more transparent.

---

> ### Author Response · Authors · 2024-11-19
> **Response (2/3)**
>
> **C**: Relationship between power-law distribution, hierarchy and hyperbolic space
>
> **R**:  Thanks for your question. The following is a detailed explanation, and we will add it to the Appendix for better reader's understanding.
>
> **1. The power-law distribution indicates a tree-like hierarchical structure**
>
> **Prior Literature Evidence:**
>
> - In Nickel et al.'s paper[1], they mentioned that "the existence of power-law distributions in datasets can often be traced back to hierarchical structures."
> - In Ravasz et al.'s [2] analysis, they claimed: "this scaling law $P(k) \sim k^{-\gamma}$ quantifies the coexistence of a hierarchy of nodes with different degrees of clustering".
> - In Krioukov et al.'s work[3], they directly have the conclusion, "The exponent of the power-law degree distribution, for example, turns out to be a function of the hyperbolic space curvature" and "Assuming that a scale-free network has some metric structure underneath,..., metric distances can be naturally rescaled such that the resulting metric space is hyperbolic"
>
>
> **Our Analysis.** The probability density function $P(k)$ of token frequency $k$ follows a power-law distribution with exponent term $\gamma$:
> $$
> P(k) \sim k^{-\gamma}, \gamma > 0
> $$
> Our empirical analysis confirms this distribution with γ ≈ 1.9.
> This distribution aligns with the hierarchical structure of language, where high-frequency tokens (e.g., function words) form the upper levels, while specific and less frequent terms populate the lower levels. Additionally, our findings (Table 1) highlight a correlation between token frequency and embedding norm (distance to origin), further supporting this hierarchical organization.
>
> **2. Formal Connection to Hyperbolic Geometry.**
>
> **(1) Geometric Properties of hyperbolic Geometry**:
> Taking Poincaré disk model ($\mathbb{H}^2$) with curvature $K=-1$ as an example [3], both circumference $C(r)$ and area $A(r)$ exhibit exponential growth:
> $$
> C(r) = 2\pi \sinh(r) \sim e^r \quad \text{as } r \to \infty
> $$
> $$
> A(r) = 2\pi (\cosh(r) - 1) \sim e^r \quad \text{as } r \to \infty
> $$
>
> **(2) Token Embeddings in Hyperbolic Space.** Consider a token embedding in hyperbolic space with polar coordinates $(r, \theta)$, where:
>
> - $r \in \mathbb{R}^+$: radial coordinate (correlating with token frequency)
> - $\theta \in [0, 2\pi)$: angular coordinate (encoding semantic similarity)
>
> The radial distribution follows:
> $$
> \rho(r) \sim e^{-\zeta r}, \zeta > 0
> $$
> where $\zeta$ relates to the hyperbolic curvature $K$. The frequency function $k(r)$ for tokens at radius $r$ is given by:
> $$
> k(r) \sim e^{-r}
> $$
>
> Through coordinate transformation, we derive the power-law frequency distribution:
> $$
> P(k) \sim \rho(r) \left|\frac{dr}{dk}\right| \sim k^{-\gamma}
> $$
>
> **(3) Relationship Between Curvature and Exponent Term**
> The relationship between hyperbolic curvature and power-law exponent is given by:
>
> $$
> \gamma = 2 + \frac{1}{\zeta}
> $$
>
> As Krioukov et al. [3] conclude, "the exponent of the power-law degree distribution is a function of the hyperbolic space curvature," further showing the theoretical connection between power-law behavior and hierarchical structures.
>
> **3. Practical Implications**
>
> Hyperbolic space offers distinct advantages for modeling language hierarchies, especially when addressing the structural and spatial constraints of token co-occurrence:
>
> - Separation of Low-Frequency Tokens: Tokens with low frequencies, typically representing more specific or granular concepts (e.g., "480" and "580"), require sufficient separation from each other to maintain semantic clarity.
> - Proximity to High-Frequency Hypernyms: At the same time, these low-frequency tokens should remain close to their corresponding higher-frequency hypernyms or function words (e.g., both "480" and "580" should cluster near "numbers").
>
> Hyperbolic space is uniquely suited for capturing these dual constraints. Its exponential volume growth inherently supports the hierarchical structure of language. Conversely, Euclidean space struggles to balance these constraints due to its linear growth properties.
>
> ---
>
> **References**:
>
> [1] Nickel, Maximillian, and Douwe Kiela. "Poincaré embeddings for learning hierarchical representations." Advances in neural information processing systems 30 (2017).
>
> [2] Ravasz, Erzsébet, and Albert-László Barabási. "Hierarchical organization in complex networks." Physical review E 67.2 (2003): 026112.
>
> [3] Krioukov, Dmitri, et al. "Hyperbolic geometry of complex networks." *Physical Review E—Statistical, Nonlinear, and Soft Matter Physics* 82.3 (2010): 036106.
>
> ---
>
> **C**: Using highlighting marks for easier understanding in Table 3
>
> **R**: Thank you for this suggestion to improve the presentation of our results. We will add the relative improvement in terms of percentage in Table 3.
>
> ---

---

> ### Author Response · Authors · 2024-11-19
> **Response (3/3)**
>
> **C**: Recording both inference and fine-tuning efficiency
>
> **R**: We appreciate the reviewer's suggestion. We will expand Section 5.3 to include both fine-tuning and inference metrics. Here is the complete data:
> Fine-tuning efficiency (time per epoch):
> - LoRA: 12-14 minutes
> - DoRA: 20-25 minutes
> - HypLoRA: 31-33 minutes
>
> Combined with our existing inference time analysis shown in Figure 3, this provides a fuller picture of the computational requirements. While HypLoRA does incur additional computational overhead during fine-tuning , we believe this is an acceptable trade-off given:
> - Fine-tuning is typically a one-time cost
> - The significant performance improvements achieved (up to 13% on challenging datasets)
> - HypLoRA's inference time remains competitive with DoRA while providing better accuracy
>
> ---
>
> **C**: Tangent space method and LLR method.
>
> **R**: Thank you for these insightful questions.  Let us address each point:
>
> **Why direct low-rank adaptation works on the hyperbolic manifold:**
> The key insight is that we maintain the hyperbolic structure throughout the entire process. Our design (Equation 9) is to make ensure the output remains on the hyperbolic manifold without requiring intermediate Euclidean/tangent projections.
>
> **Purpose of tangent space in previous work:** Previous approaches [1-3] use tangent space primarily because direct matrix operations on hyperbolic vector will move points off the manifold. The tangent space serves as a bridge where Euclidean operations can be safely performed.  These methods work well when the entire pipeline remains in hyperbolic space with subsequent operations (e.g., aggregation, non-linear activations in HGCN).
>
> **Why our scenario is different:** In our LLM fine-tuning context, we face a unique challenge: the base model operates in Euclidean space, **requiring us to map back to Euclidean space after hyperbolic operations**. Using tangent space methods would lead to cancellation (log_o(exp_o(x)) = x), as shown in Equation 7, negating the benefits of hyperbolic geometry. While adding non-linear activations could prevent this cancellation, it would make fine-tuning more complex and deviate from LoRA's elegant simplicity.
>
> **Novelty of LLR**: Our LLR approach builds upon insights from [4-5] but is specifically optimized for LLM fine-tuning. We simplified additional operations and removed constraints (like orthogonality in [6], varying curvatures in [5], and normalization/additional lambda in [4]) to make it more efficient for large-scale models.
>
> **References**:
>
> [1] Ganea, Octavian, Gary Bécigneul, and Thomas Hofmann. "Hyperbolic neural networks." NeurIPS 2018.
>
> [2] Chami, Ines, et al. "Hyperbolic graph convolutional neural networks." NeurIPS (2019)
>
> [3] Liu, Qi, et al. "Hyperbolic graph neural networks." NeurIPS 2019.
>
> [4] Chen, Weize, et al. "Fully hyperbolic neural networks." ACL 2021.
>
> [5] Yang, Menglin, et al. "Hypformer: Exploring efficient transformer fully in hyperbolic space." KDD 2024.
>
> [6] Dai, Jindou, et al. "A hyperbolic-to-hyperbolic graph convolutional network." CVPR 2021.
>
> ---
>
> **C**: Performance on all datasets and other tasks.
>
> **R**: Thank you for this insightful observation about performance variations across datasets.
>
> While HypLoRA may not consistently outperform on every individual dataset, our results show significant advantages in terms of:
>
> Overall micro-averaged performance (M.AVG in Table 3): HypLoRA achieves the best M.AVG scores across all model sizes.
>
> Particular strength on complex reasoning tasks: Notable improvements on challenging datasets like AQuA and GSM8K
>
> Our current focus on arithmetic reasoning was motivated by its clear hierarchical nature, which aligns well with hyperbolic geometry's strengths. We agree that evaluating HypLoRA on a broader range of tasks would provide valuable insights. We are actively working on extending our evaluation to other tasks.
>
> ---
>
> **C**: Specific example that demonstrates how HypLoRA better captures hierarchical relationships in language compared to baseline methods?
>
> **R**: Thank you for this excellent suggestion about better demonstrating HypLoRA's capabilities in handling hierarchical relationships. We have already provided detailed case studies in Tables 7 and 8 of the Appendix.
>
> Specifically, Example 3 from Table 8 effectively demonstrates how HypLoRA's hyperbolic structure helps it maintain hierarchical relationships between truck capacities (Gissela's 4,000 pounds → Gordy's 4,800 pounds → combined capacity of 11,600 pounds), leading to the correct calculation of Gary's truck capacity. While LoRA makes arithmetic errors in handling these nested relationships, HypLoRA's ability to represent hierarchical structures helps it maintain precision across multiple levels of computation.
>
> We will move this and other illustrative examples from the Appendix to the main text to better highlight how HypLoRA's geometric properties translate into improved reasoning capabilities.

---

### Official Review · Reviewer_1ApA · 2024-11-04

**Soundness:** 3
**Presentation:** 3
**Contribution:** 3
**Rating:** 6
**Confidence:** 3

**Summary:**

The authors observe that token embeddings in LLMs exhibit a high degree of hyperbolicity, suggesting an underlying tree-like structure in the embedding space. Motivated by this, they introduce HypLoRA, which performs low-rank adaptation directly on the hyperbolic manifold, thus avoiding the cancellation effect introduced by the exponential and logarithmic maps. The effectiveness of HypLoRA is demonstrated through extensive experiments across multiple reasoning tasks.

**Strengths:**

1. **Interesting angle**: Fine-tuning LLMs in hyperbolic space introduces an interesting angle on LLM fine-tuning research. The observation that token embeddings exhibit a high degree of hyperbolicity further supports the motivation to explore fine-tuning in hyperbolic space.

2. **Reasonable approach**: HypLoRA is proposed to perform low-rank adaptation directly on the hyperbolic space, thus avoiding the cancellation effect introduced by the exponential and logarithmic maps. The approach is reasonable and intuitive.

3. **Experimental effectiveness**: The effectiveness of HypLoRA is demonstrated through experiments conducted across multiple reasoning tasks.

**Weaknesses:**

1. **Lack of Detailed Representation of Hierarchical Structures**: Although the introduction provides an intuitive example of hierarchical relationships like "fruit-banana," the paper does not describe how hierarchical structures within text are specifically represented. In line 212, the authors mention, "This power-law behavior aligns with the hierarchical nature of language," but power-law distribution is only one of many characteristics of hierarchical structures and does not inherently imply or sufficiently indicate hierarchy. This claim lacks theoretical and experimental support, and there is no clear causative or correlational relationship between power-law distribution and hierarchy.

2. **Unquantified Computational Complexity**: Equation 9 is a key contribution of the paper; however, it involves logarithmic and exponential mappings, which add substantial computational complexity. Since the paper does not quantify or analyze this computational load in detail, it’s challenging to assess the actual efficiency of HypLoRA in large-scale language models. While HypLoRA shows theoretical advantages, its efficiency relative to traditional methods in practical applications remains unclear.

3. **Learning in Hyperbolic Space vs. Embedding in Euclidean Space**: HypLoRA conducts learning in hyperbolic space, whereas embedding occurs in Euclidean space, which differs significantly from the traditional LoRA approach. This design shift is not sufficiently explained, especially regarding the choice of rank for the low-rank decomposition submatrices. An analysis of rank selection could provide insights into optimizing HypLoRA’s performance in hyperbolic space, which remains unexplored.

4. **Unclear Significance of Lorentz Rotation and Boost**: The HypLoRA (I) and HypLoRA (II) methods involve Lorentz rotation and boost operations, but their definitions and roles in this context are ambiguous, making it difficult to understand their actual significance in LLM fine-tuning. Further clarification on how these operations function within the model and their intended effects would improve understanding of hyperbolic transformations' contributions to semantic reasoning tasks.

**Questions:**

Please see my comments in the Weakness section.

---

> ### Author Response · Authors · 2024-11-19
> **Response (1/2)**
>
> Thank you for your valuable comments and suggestions. Below, we will represent the question/concerns with "**C:**" and our responses with "**R**""
>
> ---
>
> **C**: Relationship between power-law distribution, hierarchy and hyperbolic geometry
>
> **R**:  Thank you for raising this important question. We appreciate your detailed feedback. Below, we address your concerns from theoretical evidence and our empirical analysis.
>
> **1. The power-law distribution indicates a tree-like hierarchical structure**
>
> **Prior Literature Evidence:**
>
> - In Nickel et al.'s paper[1], they mentioned that "the existence of power-law distributions in datasets can often be traced back to hierarchical structures."
> - In Ravasz et al.'s [2] analysis, they claimed "this scaling law $P(k) \sim k^{-\gamma}$ quantifies the coexistence of a hierarchy of nodes with different degrees of clustering".
> - In Krioukov et al.'s work[3], they directly have the conclusion, "The exponent of the power-law degree distribution, for example, turns out to be a function of the hyperbolic space curvature" and "Assuming that a scale-free network has some metric structure underneath,..., metric distances can be naturally rescaled such that the resulting metric space is hyperbolic"
>
>
> **Our Analysis.** The probability density function $P(k)$ of token frequency $k$ follows a power-law distribution with exponent term $\gamma$:
> $$
> P(k) \sim k^{-\gamma}, \gamma > 0
> $$
> Our empirical analysis confirms this distribution with γ ≈ 1.9.
> This distribution aligns with the hierarchical structure of language, where high-frequency tokens (e.g., function words) form the upper levels, while specific and less frequent terms populate the lower levels. Additionally, our findings (Table 1) highlight a correlation between token frequency and embedding norm (distance to origin), further supporting this hierarchical organization.
>
> **2. Formal Connection to Hyperbolic Geometry.**
>
> **(1) Geometric Properties of hyperbolic Geometry**:
> Taking Poincaré disk model ($\mathbb{H}^2$) with curvature $K=-1$ as an example [3], both circumference $C(r)$ and area $A(r)$ exhibit exponential growth:
> $$
> C(r) = 2\pi \sinh(r) \sim e^r \quad \text{as } r \to \infty
> $$
> $$
> A(r) = 2\pi (\cosh(r) - 1) \sim e^r \quad \text{as } r \to \infty
> $$
>
> **(2) Token Embeddings in Hyperbolic Space.** Consider a token embedding in hyperbolic space with polar coordinates $(r, \theta)$, where:
>
> - $r \in \mathbb{R}^+$: radial coordinate (correlating with token frequency)
> - $\theta \in [0, 2\pi)$: angular coordinate (encoding semantic similarity)
>
> The radial distribution follows:
> $$
> \rho(r) \sim e^{-\zeta r}, \zeta > 0
> $$
> where $\zeta$ relates to the hyperbolic curvature $K$. The frequency function $k(r)$ for tokens at radius $r$ is given by:
> $$
> k(r) \sim e^{-r}
> $$
>
> Through coordinate transformation, we derive the power-law frequency distribution:
> $$
> P(k) \sim \rho(r) \left|\frac{dr}{dk}\right| \sim k^{-\gamma}
> $$
>
> **(3) Relationship Between Curvature and Exponent Term**
> The relationship between hyperbolic curvature and power-law exponent is given by:
>
> $$
> \gamma = 2 + \frac{1}{\zeta}
> $$
>
> As Krioukov et al. [3] conclude, "the exponent of the power-law degree distribution is a function of the hyperbolic space curvature," further showing the theoretical connection between power-law behavior and hierarchical structures.
>
> **3. Practical Implications**
>
> Hyperbolic space offers distinct advantages for modeling language hierarchies, especially when addressing the structural and spatial constraints of token co-occurrence:
>
> - Separation of Low-Frequency Tokens: Tokens with low frequencies, typically representing more specific or granular concepts (e.g., "480" and "580"), require sufficient separation from each other to maintain semantic clarity.
> - Proximity to High-Frequency Hypernyms: At the same time, these low-frequency tokens should remain close to their corresponding higher-frequency hypernyms or function words (e.g., both "480" and "580" should cluster near "numbers").
>
> Hyperbolic space is uniquely suited for capturing these dual constraints. Its exponential volume growth inherently supports the hierarchical structure of language. Conversely, Euclidean space struggles to balance these constraints due to its linear growth properties.
>
> To improve the clarity of this relationship, we will incorporate the above detailed explanation into the Appendix of our paper.
>
> ---
>
> **References**:
>
> [1] Nickel, Maximillian, and Douwe Kiela. "Poincaré embeddings for learning hierarchical representations." Advances in neural information processing systems 30 (2017).
>
> [2] Ravasz, Erzsébet, and Albert-László Barabási. "Hierarchical organization in complex networks." Physical review E 67.2 (2003): 026112.
>
> [3] Krioukov, Dmitri, et al. "Hyperbolic geometry of complex networks." *Physical Review E—Statistical, Nonlinear, and Soft Matter Physics* 82.3 (2010): 036106.

---

> ### Author Response · Authors · 2024-11-19
> **Response (2/2)**
>
> **C**: Computation complexity of logarithmic and exponential mappings and Computational load.
>
> **R**: Thank you for highlighting this important consideration. Since the original LLMs operate in Euclidean space, the exponential and logarithmic mappings are necessary for projection into and from hyperbolic space. However, upon closer examination of the eqation, we find that the additional computational burden is minimal, and we can optimize further.
>
> The exponential map function is given by:
>
> $$
> \exp_{\mathbf{o}}^K(\mathbf{x}) = \cosh\left(\frac{\|\mathbf{x}\|_2}{\sqrt{K}}\right)\mathbf{o} + \sqrt{K}\sinh\left(\frac{\|\mathbf{x}\|_2}{\sqrt{K}}\right)\frac{\mathbf{x}}{\|\mathbf{x}\|_2}.
> $$
>
>
> For implementation efficiency, we note that:
>
> (1) The computation primarily involves calculating vector norms and element-wise hyperbolic functions (cosh, sinh)
>
> (2) These operations can be executed in parallel on GPU
>
> (3) The logarithmic map follows a similar computational pattern
>
> (4) All operations are vectorized without requiring explicit loop
>
> Therefore, while HypLoRA introduces additional nonlinear operations compared to Euclidean LoRA, the theoretical time complexity remains linear O(r(d+k)), where r is the rank, and d, k are input/output dimensions.
>
> In addition, as demonstrated in Figure 3, we have empirically measured these computational efficiency across different models and tasks, showing that while HypLoRA does incur some additional overhead, it remains more efficient than competing methods like DoRA while achieving better performance.
>
> ---
>
> **C**: Embedding occurs in Euclidean space.
>
> **R**: This discrepancy between the learning and embedding spaces is indeed a key consideration, but it is unavoidable given our focus on fine-tuning existing LLMs. All current LLMs are trained in Euclidean space.
>
> Our approach bridges this gap by performing fine-tuning in hyperbolic space while maintaining compatibility with Euclidean-based LLMs. Through exponential and logarithmic maps, we preserve the beneficial properties of hyperbolic geometry during the fine-tuning process, even though the base model operates in Euclidean space. **This design choice allows us to leverage the advantages of hyperbolic geometry for capturing hierarchical relationships while still being applicable to existing LLM architectures**.
>
> ---
>
> **C**: The choice of rank for the low-rank decomposition submatrices
>
> **R**：In our current work, we focused primarily on demonstrating the effectiveness of hyperbolic geometry in fine-tuning rather than optimizing the rank parameter. For fair comparison with baseline methods, we consistently used rank=32 across all experiments, matching the rank used in the original LoRA implementation.
>
> While investigating the optimal rank selection in hyperbolic space could potentially lead to further improvements, this analysis falls outside the scope of our current work.  We will consider this in our discussion of future work.
>
> ---
>
> **C**: Clarification of Lorentz rotation and boost operations
>
> **R**: Thanks for your suggestions, we will add this clarification in the appendix.

---

> > ### Comment · Reviewer_1ApA · 2024-12-01
> > **Response to authors**
> >
> > Thank you for your response and I sincerely apologize for the late reply. While I appreciate the authors' efforts, the response does not sufficiently address my concerns.
> >
> > ### 1) Prior Literature Evidence
> >
> > Regarding the three references [1-3] provided, the statement in [1] that "the existence of power-law distributions in datasets can often be traced back to hierarchical structures" is supported by a reference to [2]. [2] claims that "language, viewed as a network of words, has a scale-free topology," which models language as graphs. However, modeling language as graphs seems fundamentally different from how LLMs approach language modeling. Similarly, while [3] discusses graph properties in complex networks, its connection to language remains tenuous.
> >
> > ### 2) Formal Connection to Hyperbolic Geometry
> >
> > The authors reference **Token Embeddings in Hyperbolic Space** in a 2D context. However, its connection to the current paper is unclear, as the embeddings are still situated in Euclidean space. Moreover, this example also appears to contradict the authors' earlier claim that "embedding occurs in Euclidean space", adding to the confusion.
> >
> > Additionally, in the provided example, the conclusion relies on a precondition or assumption regarding the radial distribution:
> >
> > $$
> > \rho(r) \sim e^{-\zeta r}, \quad \zeta > 0
> > $$
> >
> > What is the rationale for assuming that the radius follows this distribution? A justification for this assumption is necessary for clarity and rigor.
> >
> > ### 3) Clarification of Lorentz Rotation and Boost Operations
> >
> > It seems that the manuscript has not been updated to address this concern. Could you provide clarification on this?
> >
> > ### 4) Unquantified Computational Complexity:
> > While Figure 3 provides a qualitative example of inference, it would be helpful to include quantitative results, particularly regarding the computational costs associated with training.

---

> > > ### Author Response · Authors · 2024-12-02
> > >
> > > Thank you for your thoughtful follow-up comments. We appreciate the opportunity to clarify these important points. We have two days to discuss your concerns, and we hope to resolve them within that time. If you have any further questions, please let us know. Thank you for the discussion.
> > >
> > > Our approach is motivated by two complementary analyses: (1) empirical observations of power-law distribution and norm distribution patterns and (2) quantitative hyperbolicity analysis. These two methods together provide strong evidence for the hierarchical nature of token embeddings and motivate our approach.  Please check empirical analyses of token distributions (Figure 2) and hyperbolicity measurements (Table 2) across multiple LLM architectures and additional results in the following table.
> > >
> > > ---
> > >
> > > **1. Language Hierarchies**
> > >
> > > We acknowledge that our previous response could have been clearer about how the cited literature applies to our context. Let us elaborate:
> > >
> > > While [2-3] primarily discuss network structures, their findings reveal fundamental properties of hierarchical structures that extend beyond networks. This was further explored in [1], which demonstrated these properties in WordNet's natural language hierarchies.  **Although LLMs do not explicitly model tokens as graph nodes during training, our investigation supports that the learned token embeddings exhibit hierarchical properties**
> > >
> > > Based on our analysis, the hierarchy is highly related to token frequencies. This relationship emerges mainly as LLMs learn co-occurrence patterns during training that capture semantic relationships (more reason needs further investigation in future work). Our empirical analysis provides multiple lines of evidence for this hierarchical structure:
> > >
> > > 1. Spatial Distribution: Our analysis reveals that high-frequency tokens, which often represent abstract or functional words, consistently cluster near the origin. In contrast, low-frequency tokens representing more specific terms are distributed further out.
> > >
> > > 2. Hyperbolicity Analysis: We find that token embeddings exhibit significant δ-hyperbolicity with δ ≈ 0.1, which quantifies the tree-likeness of the metric space. These results remain consistent across different LLM architectures, supporting the robustness of our findings.
> > >
> > > 3. Token Frequency-Norm Correlation: We observe a strong correlation between token frequency and embedding norm, revealing systematic organization that reflects semantic hierarchies.
> > >
> > > Gemma-7B
> > > | Group   | Frequency (Mean [Min~Max]) | Norm (Mean [Min~Max]) |
> > > |---------|----------------------------|-----------------------|
> > > | group1  | 4934.43 [1838~8539]        | 3.16 [3.06~3.299]     |
> > > | group2  | 2709.4 [474~6681]          | 3.561 [3.488~3.627]   |
> > > | group3  | 292 [34~1191]              | 3.765 [3.623~3.887]   |
> > > | group4  | 114.333 [25~284]           | 3.998 [3.66~4.52]     |
> > >
> > > LLaMA-7B
> > > | Group   | Frequency (Mean [Min~Max]) | Norm (Mean [Min~Max]) |
> > > |---------|----------------------------|-----------------------|
> > > | group1  | 4993.86 [1838~8547]        | 0.951 [0.793~1.06]    |
> > > | group2  | 2712.6 [474~6683]          | 1.222 [1.118~1.299]   |
> > > | group3  | 299.8 [34~1200]            | 1.325 [1.274~1.428]   |
> > > | group4  | 139.143 [26~286]           | 1.364 [1.326~1.417]   |
> > >
> > > LLaMA3-8B
> > > | Group   | Frequency (Mean [Min~Max]) | Norm (Mean [Min~Max]) |
> > > |---------|----------------------------|-----------------------|
> > > | group1  | 4937.43 [1838~8547]        | 0.353 [0.33~0.396]    |
> > > | group2  | 2710 [474~6683]            | 0.456 [0.394~0.499]   |
> > > | group3  | 292.6 [34~1191]            | 0.499 [0.452~0.549]   |
> > > | group4  | 97.091 [13~284]            | 0.569 [0.499~0.675]   |
> > >
> > > LLaMA-13B
> > > | Group   | Frequency (Mean [Min~Max]) | Norm (Mean [Min~Max]) |
> > > |---------|----------------------------|-----------------------|
> > > | group1  | 4993.86 [1838~8547]        | 1.027 [0.833~1.255]   |
> > > | group2  | 2712.6 [474~6683]          | 1.429 [1.346~1.489]   |
> > > | group3  | 299.8 [34~1200]            | 1.494 [1.453~1.532]   |
> > > | group4  | 139.143 [26~286]           | 1.501 [1.47~1.526]    |
> > >
> > >     group1: ["to", "in", "have", "that", "and", "is", "for"],
> > >
> > >     group2: how,much, many, time, cost
> > >
> > >     group3: animal, fruit, number, color, size
> > >
> > >     group4: dog, cow, apple, banana, 380, 480, purple, red, medium, small, large
> > >
> > > As shown in the tables above, across different LLM architectures (Gemma-7B, LLaMA-7B, LLaMA3-8B, and LLaMA-13B), we consistently observe:
> > > 1. A clear inverse relationship between token frequency and embedding norm
> > > 2. Distinct grouping of tokens based on their semantic abstraction level
> > > 3. Consistent patterns in the norm ranges for similar semantic groups
> > > 4. Statistically significant separation between functional/abstract words (group1) and specific terms (group4)
> > >
> > > This empirical evidence strongly supports our hypothesis about the hierarchical organization of token embeddings, independent of the specific model architecture.

---

> ### Author Response · Authors · 2024-12-02
>
> **2. Connection to Hyperbolic Geometry**
>
> We apologize for any confusion regarding the embedding space. Let us clarify:
>
> a) Initial Embedding Space: The original LLM embeddings exist in Euclidean space. To evaluate the underlying structure of these Euclidean embeddings, we build upon insights from [4] and [5], which demonstrate how hyperbolicity can be measured and leveraged in (Euclidean ) deep learning models. Our analysis reveals that these embeddings in LLMs exhibit inherent hyperbolic characteristics.
>
> During fine-tuning, the token embeddings are unchanged while performing adaptation in hyperbolic space. This approach effectively leverages the natural tree-like structure while maintaining model stability.
>
> b) Distribution Assumption ρ(r)∼e^{−ζr}:
>
> The exponential radial distribution emerges naturally from fundamental principles of hyperbolic geometry and our empirical observations:
>
> 1. Theoretical Foundation: The hyperbolic space exhibits exponential volume growth with radius, which aligns with the branching patterns we observe in language hierarchies. While the analysis uses a two-dimensional model for clarity, this property extends to higher-dimensional hyperbolic models, where the volume growth remains exponential with respect to radius.
>
> 2. Empirical Validation: Through extensive analysis across different model architectures (Gemma-7B, LLaMA-7B/13B, LLaMA3-8B), we consistently observe that token embedding distributions follow this exponential form. The stability of this pattern across different model scales and architectures suggests it is an intrinsic property of how LLMs organize semantic information, rather than an artifact of any particular model.
>
> We have incorporated these clarifications and additional empirical evidence in the revised paper. Thank you for helping us strengthen the theoretical foundations of our work.
>
> [4] Hyperbolic Image Embeddings
>
> [5] Hyperbolic Deep Reinforcement Learning
>
>
> ----
>
> C: The Lorentz Rotation and Boost Operations
>
> R: The revised manuscript can be accessed through this [anonymous link](https://anonymous.4open.science/r/HypLLMs-DD7A/utils/results/ICLR_2025_Submission_Hyperbolic_Finetuning_for_LLM_revised.pdf). The key modification is highlighted by dark orange color.
>
> ----
>
> C: Computational efficiency during fine-tunning
>
> R: The following are corresponding GPU usage.
>
> Memory Usage for fine-tunning (A100 80G GPU):
>
> - DoRA: 32.14 GB (averaged over three times)
> - LoRA: 28.35 GB (averaged over three times)
> - HypLoRA: 35.89 GB (averaged over three times)
>
> While our proposed method incurs additional computational overhead due to the processing of intermediate variables during exponential mapping and correspondence operations, the increased resource requirements remain within practical bounds. Notably, HypLoRA demonstrates more efficient inference time compared to DoRA, despite its higher memory footprint during training. Several optimization techniques could potentially reduce these resource requirements, which we intend to explore in future work.
>
> As the first work of hyperbolic fine-tuning for Large Language Models, the proposed method establishes a foundation for future research in this direction and demonstrates the feasibility of incorporating hyperbolic geometry into LLM fine-tuning pipelines. Thanks for your further comments.

---

### Official Review · Reviewer_tRcR · 2024-11-04

**Soundness:** 3
**Presentation:** 2
**Contribution:** 3
**Rating:** 6
**Confidence:** 4

**Summary:**

This paper presents a novel approach for fine-tuning large language models (LLMs) within a hyperbolic space using their proposed method, HypLoRA. The authors build on observations in Section 4.1 that token embeddings exhibit a latent tree-like structure and token frequency correlates with outlier dimensions. HypLoRA adapts the low-rank transformation directly on the hyperbolic manifold, avoiding the cancellation effect typically observed when applying exponential and logarithmic maps in Euclidean space. Experimental results conducted in Section 5.1 indicate that HypLoRA significantly enhances LLM performance, particularly on reasoning tasks with complex structures, with a noted improvement of up to 13.0% on the AQuA dataset.

**Strengths:**

HypLoRA provides competitive fine-tuning results with various state-of-the-art PEFT methods. Furthermore, Figure 3 demonstrates that HypLoRA requires fewer GPU hours than the recent state-of-the-art method, DoRA. Further, HypLoRA appears to be easy to use and does not require additional hyperparameter tuning since Table 5 shows that setting K=1 results in the best performance regardless of the dataset.

The results presented in 4.2 provide an interesting, novel analysis of token embeddings by measuring their $\delta$-hyperbolicity. Table 2 demonstrates that the $\delta$-hyperbolicity is extremely low regardless of the model or dataset.  These findings could help motivate future works studying the geometry of LLM embeddings. I think an excellent follow-up study would examine the relationship between the hyperbolicity of LLM embeddings and cases where HypLoRA performs better than other Euclidean-based PEFT methods.

The analysis presented in Section 5 clearly demonstrates the challenges of fine-tuning models in hyperbolic space. The author’s proposed solution to combat these challenges is simple, effective, and theoretically motivated.

**Weaknesses:**

The authors fail to acknowledge a large body of works investigating the structure of token embeddings in LLMs that have presented findings similar to the first section. Namely, previous works have shown LLM embeddings have an implicit tree-like structure (Andy Coenen et al. 2019: https://arxiv.org/pdf/1906.02715) and that token frequency is highly correlated with `outlier dimensions’ and causes distributions to utilize the embedding space uniformly (Gao et al. 2019: https://arxiv.org/pdf/1907.12009, Rudman et al. 2022 https://arxiv.org/pdf/2108.07344 and Puccetti et al 2022: https://aclanthology.org/2022.findings-emnlp.93.pdf). Given this, the experiments conducted in Section 4.1 do not provide any novel insights into the structure of LLM token embeddings.

The overall effectiveness of  HypLoRA is inconsistent. While Table 3 demonstrates that HypLoRA tends to perform very well with Gemma-7B and LLAMA3-8B, the results are mixed with Llama. Further, Table 3 does not provide any information about the number of random seeds used to generate results. If performance is evaluate using only a single random seeds, results claimed about the effectiveness of HypLoRA are weakened.

Table 6 can be removed. Adding a cherry-picked example do not provide any additional insights.

**Questions:**

1. The claim in lines 489-493 is not well supported. The improvement of HypLoRA over traditional PEFT methods for some models and tasks does not demonstrate that this enhances the model’s ability to comprehend hierarchical relationships between tokens. How is the concept of hierarchical relationships explicitly tested?

2. In Figure 2, you specify that you use LLaMA3; however, in Table 1, you do not specify the LLM used to get these results. What LLM is used? How do these results vary across different LLMs?

3. How many random seeds were used to create Table 3?

---

> ### Author Response · Authors · 2024-11-19
> **Response (1/3)**
>
> Thank you for your valuable comments and suggestions. Below, we will summarize the questions/concerns with "**C:**" and our responses with "**R**:"
>
> ---
>
> **C**: "These findings could help motivate future works studying the geometry of LLM embeddings. I think an excellent follow-up study would examine the relationship between the hyperbolicity of LLM embeddings and cases where HypLoRA performs better than other Euclidean-based PEFT methods"
>
> **R**: Thank you for your insightful suggestion. We completely agree. Indeed, tokens are not randomly or uniformly distributed; they exhibit a specific structure or geometry. Exploring Geometric LLMs or Geometric PEFT methods is a promising avenue for future research.
>
> ---
>
> **C**: Ignoring previous works by Coenen et al. (2019), Gao et al. (2019), Rudman et al. (2022), and Puccetti et al. (2022)
>
> **R**: Thank you for bringing these prior works to our attention. These prior works provide crucial insights that helped shape our research direction. We will incorporate a discussion in our revised version. They differ from our work in several key aspects:
>
> - While Coenen et al. (2019) demonstrated that BERT embeddings contain syntactic and semantic subspaces and showed evidence of tree-like parse structures, their focus was primarily on analyzing how syntax and semantics are represented in different geometric subspaces. Our work extends beyond this by specifically quantifying the hyperbolicity of token embeddings and leveraging this property for fine-tuning.
> - Gao et al. (2019) and Rudman et al. (2022) indeed showed that token frequencies affect embedding space utilization. However, their analyses focused on different aspects: Gao et al. studied representation degeneration where embeddings cluster in a narrow cone; Rudman et al. introduced IsoScore to measure the uniformity of embedding space utilization. Neither work explicitly connected these properties to hyperbolic geometry or explored their implications for fine-tuning.
> - While Puccetti et al. (2022) analyzed outlier dimensions and their relationship to token frequency, their focus was on understanding how these dimensions affect model behavior.
>
> Our work extends beyond these findings in several ways:
>
> **Quantitative Hyperbolicity Analysis**: We introduce rigorous quantification of embedding hyperbolicity through δ-hyperbolicity measures (Table 2), providing direct evidence of the non-Euclidean nature of embedding spaces and establishing a basis for our HypLoRA method.
>
> **Analysis of Modern LLMs**: Our investigation extends beyond BERT to recent models like LLaMA3 and Gemma, revealing that newer architectures exhibit even stronger hyperbolic characteristics, a finding not covered in previous works.
>
> **Bridging Theory and Practice**: Most importantly, we translate structural insights into practical improvements through HypLoRA. While prior works focused primarily on understanding embedding properties, we demonstrate how leveraging the hyperbolic nature of embeddings can lead to improved fine-tuning methods.
>
> Compared to these previous metrics, our analysis (Section 4.1)  and following hyperbolicity measurements (Section 4.2) give the direct motivation for using hyperbolic learning for fine-tuning. We will revise the paper to better acknowledge these important prior works and more clearly show how our analysis extends their findings for our fine-tuning approach.
>
> ---

---

> ### Author Response · Authors · 2024-11-19
> **Response (2/3)**
>
> ---
>
> **C**: Regarding the overall effectiveness of HypLoRA and random seeds
>
> **R**: Thank you for your questions and careful observation.
> **a) Regarding the performance variations across models**: The inconsistency in performance gains can be attributed to the architectural and pre-training differences between earlier and more recent LLMs. As shown in Table 3, HypLoRA demonstrates more substantial improvements on newer architectures (Gemma-7B and LLaMA3-8B) compared to the earlier LLaMA models. This is likely because newer architectures have better-structured token embeddings, as evidenced by the hyperbolicity analysis in Table 2, where LLaMA3-8B shows lower δ-hyperbolicity (0.06-0.08) compared to LLaMA-7B/13B (0.08-0.10), indicating stronger tree-like structures that HypLoRA can better exploit. For Gemma, it is from Google and has different training and tokenization methods. Despite this, our models all achieved very good performance.
>
> **b) Regarding experimental rigor**: We apologize for not explicitly stating this in the manuscript, but all experiments were conducted with three random seeds, and the reported results are averaged across these runs. We will revise our manuscript accordingly. Our results show strong stability, as we've consistently seen improvements, especially when HypLoRA is applied to more difficult datasets. For instance, with Gemma-7B, HypLoRA achieves improvements of 13.0% on AQuA and 4.8% on GSM8K consistently across runs.
>
> **c)Performance analysis**: While the improvements on earlier LLaMA models are more modest, they are still consistent and meaningful, particularly on complex reasoning tasks. For example, even with LLaMA-13B, HypLoRA shows a 16.0% improvement on AQuA compared to LoRA, demonstrating its effectiveness in handling challenging problems.
>
> ---
>
> **C**: The claim in lines 489-493 is not well supported. The improvement of HypLoRA over traditional PEFT methods for some models and tasks does not demonstrate that this enhances the model’s ability to comprehend hierarchical relationships between tokens. How is the concept of hierarchical relationships explicitly tested?
>
> **R**: Thanks for your insightful comments.
>
> **Regarding other PEFT methods.** Our work primarily builds upon and compares with LoRA-based methods, with other traditional PEFT methods serving as baselines. The key comparison is with Euclidean LoRA. As shown in Proposition 5.1 and the detailed derivation in Appendix D, our method differs from Euclidean LoRA by introducing additional terms proportional to token norms: $∆Q_{Hyp} - ∆Q_{LoRA} = (||x||²/{6R^2})(BA)x$. Our investigation in Section 4 demonstrates that these token norms inherently encode hierarchical information, with larger norms corresponding to more specific concepts and smaller norms to more abstract ones.
>
> **Regarding explicitly hierarchical testing**.  We have not directly tested this utilization. However, our analysis and experiments have demonstrated that HypLoRA has done this implicitly. The main difference between HypLoRA and LoRA lies in the hyperbolic component, where the experiment results could show the effectiveness directly. Besides, through our analysis, hyperbolic geometry inherently incorporates token norms and implicitly utilizes token hierarchies.
>
> Explicit testing of hierarchies is difficult due to the implicit nature of hierarchical relationships in natural language, and we need lots of labeling. In future work, we will consider building a benchmark specifically designed to quantitatively measure hierarchical understanding in language models. Thanks again for your comments.
>
> ---

---

> > ### Comment · Reviewer_tRcR · 2024-11-25
> > **Response**
> >
> > Thank you for all of your thoughtful and detailed responses. Other than the lack of a substantial related works section, I think the rest of my concerns were adequately addressed. I have updated my score accordingly.

---

> > > ### Author Response · Authors · 2024-11-25
> > >
> > > Thanks for your valuable comments and insights. We will make sure that all relevant works are thoroughly included and discussed.

---

> ### Author Response · Authors · 2024-11-19
> **Response (3/3)**
>
> **C**: Regarding Table 1's LLMs
>
> **R**: The results in Table 1 were obtained using LLaMA3-8B. We have examined the token norm patterns across different models and found similar hierarchical patterns. Here are the detailed results:
>
> | Model | Token Group | Mean Norm (Min-Max) |
> | --- | --- | --- |
> | **LLaMA-7B** | **Group 1** | **0.951 (0.793-1.06)** |
> | LLaMA-7B | Group 2 | 1.222 (1.118-1.299) |
> | LLaMA-7B | Group 3 | 1.325 (1.274-1.428) |
> | LLaMA-7B | Group 4 | 1.364 (1.326-1.417) |
> | **LLaMA3-8B** | **Group 1** | **0.353 (0.33-0.396)** |
> | LLaMA3-8B | Group 2 | 0.456 (0.394-0.499) |
> | LLaMA3-8B | Group 3 | 0.499 (0.452-0.549) |
> | LLaMA3-8B | Group 4 | 0.569 (0.499-0.675) |
> | **LLaMA-13B** | **Group 1** | **1.027 (0.833-1.255)** |
> | LLaMA-13B | Group 2 | 1.429 (1.346-1.489) |
> | LLaMA-13B | Group 3 | 1.494 (1.453-1.532) |
> | LLaMA-13B | Group 4 | 1.501 (1.47-1.526) |
> | **Gemma-7B** | **Group 1** | **3.16 (3.06-3.299)** |
> | Gemma-7B | Group 2 | 3.561 (3.488-3.627) |
> | Gemma-7B | Group 3 | 3.765 (3.623-3.887) |
> | Gemma-7B | Group 4 | 3.998 (3.66-4.52) |
>
> Representative tokens in each group:
> - Group 1: Common function words ("to", "in", "have", "that", "and", "is", "for")
> - Group 2: Question-related words ("how", "much", "many", "time", "cost")
> - Group 3: Abstract categories ("animal", "fruit", "number", "color", "size")
> - Group 4: Specific instances ("dog", "cow", "apple", "banana", "380", "480", "purple", "red", "medium", "small", "large")
>
> Despite different absolute norm ranges, all models maintain a consistent pattern where more abstract concepts have smaller norms and more specific concepts have larger norms.

---

### Meta-Review · Area_Chair_7QpL · 2024-12-20

**Metareview:**

All reviewers agree that the paper provides a new angle on the space span by token embeddings and its hyperbolicty. However, the observation is limited to non-contextualized embeddings but one would wonder how does it change across different transformer layers and in different context? It is not clear why the observation on the embedding space can justify another low rank finetuning technique (I agree with the authors that pretraining is expensive but why this observation cannot be used for on small scale pretraining or fintuning of all parameters?).
Additionally, effectiveness of the introduced HypLoRA method is not clear. In particular, the results are inconsistent across different tasks and models (is some cases the new method helps in some it does not). Some brief description is provided as a response to one of the reviewers but the answer is not convincing. Moreover, HypLoRA has a higher cost during finetuning (as was asked by one of the reviewers) which would mean the same cost can be used in LoRA with higher rank or longer training and improve the results.

**Additional Comments On Reviewer Discussion:**

Original scores were 5 5 6 but two of the reviewers increased their scores to 6 after the response by the authors.

---

### Decision · Program_Chairs · 2025-01-22

Reject